# Cooperative Hardware-Prompt Learning for Snapshot Compressive Imaging

**Jiamian Wang**[1*]**, Zongliang Wu**[2,3]**, Yulun Zhang**[4]**, Xin Yuan**[2]**, Tao Lin**[2]**, Zhiqiang Tao**[1*]
[1]Rochester Institute of Technology, [2]Westlake University,
[3]Zhejiang University, [4]Shanghai Jiao Tong University

## Abstract

Existing reconstruction models in snapshot compressive imaging systems (SCI) are trained with a single well-calibrated hardware instance, making their performance vulnerable to hardware shifts and limited in adapting to multiple hardware configurations. To facilitate cross-hardware learning, previous efforts attempt to directly collect multi-hardware data and perform centralized training, which is impractical due to severe user data privacy concerns and hardware heterogeneity across different platforms/institutions. In this study, we explicitly consider data privacy and heterogeneity in cooperatively optimizing SCI systems by proposing a Federated Hardware-Prompt learning (FedHP) framework. Rather than mitigating the client drift by rectifying the gradients, which only takes effect on the learning manifold but fails to solve the heterogeneity rooted in the input data space, FedHP learns a hardware-conditioned prompter to align inconsistent data distribution across clients, serving as an indicator of the data inconsistency among different hardware (e.g., coded apertures). Extensive experimental results demonstrate that the proposed FedHP coordinates the pre-trained model to multiple hardware configurations, outperforming prevalent FL frameworks for 0.35dB under challenging heterogeneous settings. Moreover, a Snapshot Spectral Heterogeneous Dataset has been built upon multiple practical SCI systems. Data and code are aveilable at https://github.com/Jiamian-Wang/FedHP-Snapshot-Compressive-Imaging.git

## 1 Introduction

The technology of snapshot compressive imaging (SCI) [Yuan et al., 2021] has gained prominence in the realm of computational imaging. Taking an example of hyperspectral image reconstruction, the spectral SCI [Gehm et al., 2007] can fast capture and compress 3D hyperspectral signals as 2D measurements through optical hardware, and then restore the original signals with high fidelity by training deep neural networks [Meng et al., 2020, Miao et al., 2019]. Despite the remarkable performance [Cai et al., 2022a,b, Lin et al., 2022, Huang et al., 2021, Hu et al., 2022], existing deep SCI methods are generally trained with a specific hardware configuration, *e.g.*, a well-calibrated coded aperture (physical mask). The resulting model is vulnerable to hardware shift/perturbation and limited in adapting to multiple hardware configurations. However, directly learning a reconstruction model cooperatively from multi-hardware seems to be infeasible due to data proprietary constraint. It is also non-trivial to coordinate heterogeneous hardware instances with a unified model.

To elaborate, we first recap previous research efforts of centralized learning solutions. A naive solution is to *jointly train* a single reconstruction model with data collected from different hardware configurations, *i.e.*, coded apertures. As shown in Fig. 1 *right*, this solution enhances the ability of reconstruction (0.5dB+) by comparison to a single hardware training scenario. However, the

---

*Corresponding authors: Jiamian Wang (`jw4905@rit.edu`) and Zhiqiang Tao (`zhiqiang.tao@rit.edu`)

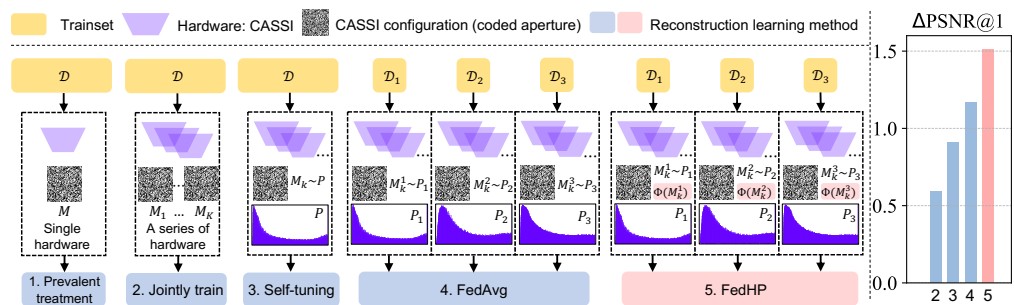

Figure 1: Comparison of hyperspectral reconstruction learning strategies. (1) The model trained with the single hardware (*Prevalent treatment*) hardly handles other hardware. Both (2) *Jointly train* and (3) *Self-tuning* [Wang et al., 2022] are centralized training solutions. Both (4) FedAvg and the proposed (5) FedHP adopt the same data split setting. We compare the performance gain of different methods over (1). All results are evaluated by unseen masks (non-overlapping) sampled from the practical mask distributions $\{P_1, P_2, P_3\}$. FedHP learns a prompt network $\Phi(\cdot)$ for cooperation.

performance on inconsistent coded apertures is still non-guaranteed since the model only learns to fit coded apertures in a purely data-driven manner. Followed by, *self-tuning* [Wang et al., 2022] advances the learning by approximating the posterior distribution of coded apertures in a variational Bayesian framework. Despite the significant performance boost, it is only compatible with the coded apertures drawing from *homogeneous* hardware (same distribution) yet cannot handle *heterogeneous* hardware. Nevertheless, centralized learning presumes that hardware instances and hyperspectral data are always publicly available, which hardly holds in practice – both the optical systems (with different confidential configurations, *e.g.*, coded apertures) and data samples (*i.e.*, measurements captured from non-overlapping scenes) are generally proprietary assets across institutions, adhering to the strict privacy policy constraints [Vergara-Laurens et al., 2016, Li et al., 2021], while considering the multi-hardware cooperative training confining to this concern remains unexplored.

In this work, we leverage federated learning (FL) [Kairouz et al., 2021, Li et al., 2020a, Wang et al., 2021] for cross-platform/silo multi-hardware reconstruction modeling without sharing the hardware configurations and local training data. Firstly, the FL benchmark, FedAvg [McMahan et al., 2017], is adopted and brings performance boost (compared by 3 and 4 in Fig. 1 *right*). However, FedAvg has been proven to be limited in solving heterogeneous data [Hsu et al., 2019, Karimireddy et al., 2020] – the heterogeneity in SCI substantially stems from the hardware, which is usually absorbed into the compressed data and governs the network training. Thus, different configurations, *e.g.*, coded apertures, yield different data distributions. Besides, we consider a more practical scenario by extending the sample-wise hardware difference into distribution-wise, *i.e.*, not only the different coded apertures yield heterogeneity, but also coded apertures from different clients may follow different distributions (see $P_1 \sim P_3$ in Fig. 1).

To adress the heterogeneity issue, this work proposes a Federated Hardware-Prompt (FedHP) framework to achieve multi-hardware cooperative learning with privacy piratically preserved. Prevalent FL methods handle the heterogeneity by regularizing the global/local gradients [Karimireddy et al., 2020, Li et al., 2020b], which only take effect on the learning manifold but fail to solve the heterogeneity rooted in the input data space. Differently, FedHP traces back to the source of the data heterogeneity of this application, *i.e.*, inconsistent hardware configurations, and devises a prompt network to solve the client drift issue in input data space. By taking the coded aperture as input, the prompter better accounts for the underlying inconsistency and closes the gap between input data distributions across clients. Besides, the prompter explicitly models the correlation between the software and hardware, empowering the learning by following the spirit of the co-optimization [Goudreault et al., 2023, Zheng et al., 2021, Robidoux et al., 2021] in computational imaging. In addition, FedHP directly operates on pre-trained reconstruction backbones with locally well-trained models and keeps them frozen throughout the learning, which improves the training efficiency than directly optimizing the reconstruction backbones in FL from scratch. We summarize the contributions as follows.

- We introduce and tackle an unexplored problem of hardware cooperative learning in SCI, under the presence of data privacy constraints and heterogeneous configurations. To our best knowledge, the proposed FedHP first integrates federated learning into spectral SCI.

- We uncover the data heterogeneity of SCI that stems from distinct hardware configurations. A hardware prompt module is developed to solve the distribution shift across clients and empower the hardware-software co-optimization in computational imaging. The proposed method provides an orthogonal perspective in handling the heterogeneity of the existing FL practices.
- We build a new Snapshot Spectral Heterogeneous Dataset (SSHD) from multiple practical spectral snapshot imaging systems. Extensive experiments demonstrate that FedHP outperforms both centralized learning methods and classic federated learning frameworks. The proposed method can inspire future work in this novel research direction of hardware collaboration in SCI.

## 2 Method

### 2.1 Preliminary Knowledge

We study the cooperative learning problem by taking the representative setup of coded aperture snapshot spectral imaging system for hyperspectral imaging as an example, due to its recent advances [Cai et al., 2022a,b, Lin et al., 2022]. Given the real-world hyperspectral signal $\mathbf{X} \in \mathbb{R}^{H \times W \times N_\lambda}$, where $N_\lambda$ denotes the number of spectral channels, the hardware performs the compression with the physical coded apterture $\mathbf{M}$ of the size $H \times W$, i.e., $\mathbf{M}_{hw} \in [0, 1]$. Accordingly, the encoding process produces a 2D measurement $\mathbf{Y}^{\mathbf{M}} \in \mathbb{R}^{H \times (W+\Delta)}$, where $\Delta$ denotes the shifting

$$\mathbf{Y}^{\mathbf{M}} = \sum_{n_\lambda=1}^{N_\lambda} \mathbf{X}'(:,:,n_\lambda) \odot \mathbf{M} + \mathbf{\Omega},$$

$$\mathbf{X}'(h, w, n_\lambda) = \mathbf{X}(h, w + d(\lambda - \lambda^*), n_\lambda),$$

(1)

where $\odot$ denotes the pixel-wise multiplication and $\mathbf{\Omega}$ presents the measurement noise. For each spectral wavelength $\lambda$, the corresponding signal $\mathbf{X}(:,:,n_\lambda)$ is shifted according to the function $d(\lambda - \lambda^*)$ by referring to the pre-defined anchor wavelength $\lambda^*$, such that $\Delta = d(N_\lambda - 1)$. Following the optical encoder, recent practices train a deep reconstruction network $f(\cdot)$ to retrieve the hyperspectral data $\widehat{\mathbf{X}} \in \mathbb{R}^{H \times W \times N_\lambda}$ by taking the 2D measurement $\mathbf{Y}^{\mathbf{M}}$ as input. We define the initial training dataset as $\mathcal{D}$ and the corresponding dataset for the reconstruction as $\mathcal{D}^{\mathcal{M}^*}$

$$\mathcal{D} = \{\mathbf{X}_i\}_{i=1}^{i=N}, \quad \mathcal{D}^{\mathbf{M}^*} = \{\mathbf{Y}_i^{\mathbf{M}^*}, \mathbf{X}_i\}_{i=1}^{i=N},$$

(2)

where $\mathbf{X}_i$ is the ground truth and $\mathbf{Y}_i^{\mathbf{M}^*}$ is governed by a specific coded aperture $\mathbf{M}^*$. The reconstruction model finds the local optimum by minimizing the mean squared loss

$$\widehat{\theta} = \arg\min_\theta \frac{1}{N} \sum_{i=1}^{N} ||f(\theta; \mathbf{Y}_i^{\mathbf{M}^*}) - \mathbf{X}_i||_2^2,$$

(3)

where $\theta$ expresses all learnable parameters in the reconstruction model. $\widehat{\mathbf{X}}_i = f(\widehat{\theta}; \mathbf{Y}_i^{\mathbf{M}^*})$ is the prediction. Pre-trained reconstruction models [Cai et al., 2022a, Huang et al., 2021] demonstrates promising performance when is compatible with a single encoder set-up, where the measurement in training and testing phases are produced by the same hardware using a fixed coded aperture of $\mathbf{M}^*$.

**Motivation**. Previous work [Wang et al., 2022] uncovered that most existing reconstruction models experience large performance descent (e.g., $> 2$dB in terms of PSNR) when handling the data encoded by a different coded aperture $\mathbf{M}^\dagger$ from training, i.e., $\mathbf{M}^\dagger \neq \mathbf{M}^*$ as mask determines the data distribution and also takes effect in learning as (3). Thus, a well-trained reconstruction model can be highly sensitive to a specific hardware configuration of coded aperture and is hardly compatible with the other optical systems in the testing phase. A simple solution of adapting the reconstruction network to a different coded aperture $\mathbf{M}^\dagger$ is to retrain the model with corresponding dataset $\mathcal{D}^{\mathbf{M}^\dagger} = \{\mathbf{Y}_i^{\mathbf{M}^\dagger}, \mathbf{X}_i\}_{i=1}^{i=N}$ and then test upon $\mathbf{M}^\dagger$ accordingly. However, this solution does not broaden the adaptability of reconstruction models to multi-hardware and can introduce drastic computation overhead. In this work, we tackle this challenge by learning a reconstruction model cooperatively from multiple hardware with inconsistent configurations.

### 2.2 Centralized Learning in SCI

**Jointly Train**. To solve the above problem, *Jointly train* (Fig. 1 part 2) serves as a naive solution to train a model with data jointly collected upon a series of hardware. Assuming there are total number

of $K$ hardware with different coded apertures, *i.e.*, $\mathbf{M}_1, \mathbf{M}_2, ..., \mathbf{M}_K$. Each hardware produces a training dataset upon $\mathcal{D}$ as $\mathcal{D}^{\mathbf{M}_k} = \{\mathbf{Y}_i^{\mathbf{M}_k}, \mathbf{X}_i\}_{i=1}^{i=N}$. The joint training dataset for reconstruction is

$$\mathcal{D}^{\mathbf{M}_{1 \sim K}} = \mathcal{D}^{\mathbf{M}_1} \cup \mathcal{D}^{\mathbf{M}_2} \cup \ldots \cup \mathcal{D}^{\mathbf{M}_K}, \tag{4}$$

where different coded apertures can be regarded as hardware-driven data augmentation treatments toward the hyperspectral data. The reconstruction model will be trained with the same mean squared loss provided in (3) upon $\mathcal{D}^{\mathbf{M}_{1 \sim K}}$. [Wang et al., 2022] demonstrated that jointly learning brings performance boost compared with single mask training (Fig. 1 *right*). However, this method adopts a single well-trained model to handle coded apertures, failing to adaptively cope with the underlying discrepancies and thus, leading to compromised performances for different hardware.

**Self-tuning**. Following *Jointly train*, recent work of *Self-tuning* [Wang et al., 2022] recognizes the coded aperture that plays the role of hyperprameter of the reconstruction network, and develops a hyper-net to explicitly model the posterior distribution of the coded aperture by observing $\mathcal{D}^{\mathbf{M}_{1 \sim K}}$. Specifically, the hyper-net $h(\sigma; \mathbf{M}_k)$ approximates $P(\mathbf{M}|\mathcal{D}^{\mathbf{M}_{1 \sim K}})$ by minimizing the Kullback–Leibler divergence between this posterior and a variational distribution $Q(\mathbf{M})$ parameterized by $\sigma$. Compared with *Jointly train*, *Self-tuning* learns to adapt to different coded apertures and appropriately calibrates the reconstruction network during training, even if there are unseen coded apertures. However, the variational Bayesian learning poses a strict distribution constraint to the sampled coded apertures, which limits the scope of *Self-tuning* under the practical setting.

To sum up, both of the *Jointly train* and *Self-tuning* are representative solutions of centralized learning, where the dataset $\mathcal{D}$ and hardware instances with $\mathbf{M}_1, ..., \mathbf{M}_K$ from different sources are presumed to be publicly available. Such a setting has two-fold limitations. (1) Centralized learning does not take the privacy concern into consideration. Hardware configuration and data information sharing across institutions is subject to the rigorous policy constraint. (2) Existing centralized learning methods mainly consider the scenario where coded apertures are sampled from the same distribution, *i.e.*, hardware origin from the same source, which is problematic when it comes to the coded aperture distribution inconsistency especially in the cross-silo case. Bearing the above challenges, in the following, we resort to the federated learning (FL) methods to solve the cooperative learning of reconstruction considering the privacy and hardware configuration inconsistency.

## 2.3 Federated Learning in SCI

**FedAvg**. We firstly tailor FedAvg [McMahan et al., 2017], into SCI. Specifically, we exploit a practical setting of cross-silo learning in snapshot compressive imaging. Suppose there are $C$ clients, where each client is packaged with a group of hardware following a specific distribution of $P_c$

$$\mathbf{M}_k^c \sim P_c, \tag{5}$$

where $\mathbf{M}_k^c$ represents $k$-th sampled coded aperture in $c$-th client. For simplicity, we use $\mathbf{M}^c$ to denote arbitrary coded aperture sample in $c$-th client as shown in Eq. (5). Based on the hardware, each client computes a paired dataset $\mathcal{D}^{\mathbf{M}^c}$ from the local hyperspectral dataset $\mathcal{D}_c$

$$\mathcal{D}_c = \{\mathbf{X}_i\}_{i=1}^{i=N_c}, \quad \mathcal{D}^{\mathbf{M}^c} = \{\mathbf{Y}_i^{\mathbf{M}^c}, \mathbf{X}_i\}_{i=1}^{i=N_c}, \tag{6}$$

where $N_c$ represents the number of hyperspectral data in $\mathcal{D}_c$. The local learning objective is

$$\ell_c(\theta) = \frac{1}{N} \sum_{i=1}^N ||\widehat{\mathbf{X}}_i - \mathbf{X}_i||_2^2, \tag{7}$$

where $\widehat{\mathbf{X}}_i = f(\widehat{\theta}; \mathbf{Y}_i^{\mathbf{M}^c})$, $\mathbf{M}^c \sim P_c$, we use $\theta$ to denote the learnable parameters of reconstruction model at a client. FedAvg learns a global model $\theta_G$ without sharing the hyperspectral signal dataset $\mathcal{D}_c, \mathcal{D}^{\mathbf{M}^c}$, and $\mathbf{M}^c$ across different clients. Specifically, the global learning objective $\ell_G(\theta)$ is

$$\ell_G(\theta) = \sum_{c=1}^{C'} \alpha_c \ell_c(\theta), \tag{8}$$

where $C'$ denotes the number of clients that participate in the current global round and $\alpha_c$ represents the aggregation weight. Compared with the centralized learning solutions, FedAvg not only bridges the local hyperspectral data without sharing sensitive information, but also collaborates multi-hardware with a unified reconstruction model for a better performance (Fig. 1 *right* comparison between 3 and 4). However, FedAvg shows limitations in two-folds. (1) It has been shown that

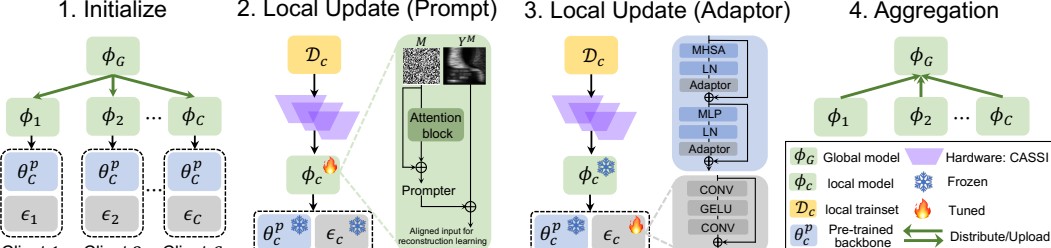

Figure 2: Learning process of FedHP. We take one global round as an example, which consists of (1) *Initialize*, (2) *Local Update (Prompt)*, (3) *Local Update (Adaptor)*, and (4) *Aggregation*. For each client, the reconstruction backbone ($\theta_c^p$), is initialized as pre-trained model upon local training dataset $\mathcal{D}_c$ and kept as frozen throughout the training. The prompt net upon hardware configuration, *i.e.*, coded aperture, takes effect on the input data of reconstruction, *i.e.*, $\mathbf{Y}^{\mathbf{M}}$. Adaptors are introduced to enhance the learning, where $\epsilon_c$ denotes the parameters of all adaptors.

FedAvg is hard to handle the heterogeneous data [Karimireddy et al., 2020, Khaled et al., 2020, Hsu et al., 2019]. (2) Directly training the reconstruction backbones from scratch would introduce prohibitive computation. Next, we firstly introduce the hardware-induced data heterogeneity in SCI. Then we develop a Federated Hardware-Prompt (FedHP) method to achieve cooperative learning without optimizing the client backbones.

**Data Heterogeneity**. We firstly consider the data heterogeneity stems from the *different coded apertures samples*, *i.e.*, hardware instances. According to Section 2.1, the optical hardware samples the hyperspectral signal $\mathbf{X}_i$ from $\mathcal{D} = \{\mathbf{X}_i\}_{i=1}^{i=N}$ and encodes it into a 2D measurement $\mathbf{Y}_i^{\mathbf{M}}$, which constitutes $\mathcal{D}^{\mathbf{M}}$ and further serves as the input data for the reconstruction model. To this end, the modality of $\{\mathbf{Y}_i^{\mathbf{M}}\}_{i=N}^{i=1}$ is vulnerable to the coded aperture variation. A single coded aperture $\mathbf{M}$ defines a unique input data distribution for the reconstruction, *i.e.*, $\mathbf{Y}_i^{\mathbf{M}} \sim P_{\mathbf{M}}(\mathbf{Y}_i^{\mathbf{M}})$. For arbitrary distinct coded apertures, we have $P_{\mathbf{M}^*}(\mathbf{Y}_i^{\mathbf{M}^*}) \neq P_{\mathbf{M}^\dagger}(\mathbf{Y}_i^{\mathbf{M}^\dagger})$ if $\mathbf{M}^* \neq \mathbf{M}^\dagger$. In federated learning, data heterogeneity persistently exists since there is no identical coded aperture across different clients. Such a heterogeneous scenario, *i.e.*, sampling *non-overlapping masks* from the same mask distribution, can be caused by lightning distortion or optical platform fluttering.

We take a step further to consider the other type of data heterogeneity stemming from the *distinct distributions of coded apertures* [2]. As formulated in (6), each client collects a coded aperture assemble following the distribution $P_c$ for $c$-th client. We have $P_c$ differs from one another, *i.e.*, $P_{c1} \neq P_{c2}$ for $c1 \neq c2, c1, c2 \in \{1, ..., C\}$. Hardware instances from different clients are produced by distinct manufacturing agencies, so that the distribution $P_{c1}$ and $P_{c2}$ drastically differs as demonstrated in Fig. 1. This is a more challenging scenario than previous case. As presented in Section 3.2, classic federated learning methods, *e.g.*, FedProx [Li et al., 2020b] and SCAFFOLD [Karimireddy et al., 2020] hardly converge while the proposed method enables an obvious performance boost.

## 2.4 FedHP: Federated Hardware-Prompt Learning

**Hardware-Prompt Learning**. Bearing the heterogeneous issue, previous efforts [Li et al., 2020b, Karimireddy et al., 2020] mainly focus on rectifying the global/local gradients upon training, which only *takes effect on the learning manifold* but fail to *solve the heterogeneity* rooted in the input data space, whose effectiveness in this low-level vision task may be limited. Since we uncover two types of the heterogeneity in snapshot compressive imaging stemming from the hardware inconsistency (Section. 2.3), this work opts to tackling the client drift issue by directly operating in the input data space. This can be achieved by collaboratively learning the input data alignment given different coded apertures. In light of the visual prompt tuning in large models [Liu et al., 2023b, Bahng et al., 2022], we devise a hardware-conditioned prompt network in the following.

As shown in the *Step 2* of Fig. 2, given the input data $\{\mathbf{Y}_i^{\mathbf{M}}\}_{i=1}^{i=N}$ of the reconstruction, the prompt network aligns the input samples, *i.e.*, measurements $\mathbf{Y}^{\mathbf{M}_i}$, by adding a prompter conditioned on the hardware configuration. Let $\Phi(\phi; \mathbf{M})$ denote the prompt network (*e.g.*, attention block) parameterized

---

[2]We presume that the hyperspectral single dataset $\mathcal{D}_c, c = 1, ..., C$, shares the same distribution by generally capturing the natural scenes. Heterogeneity stems from the hyperspectral signal is out of the scope of this work.

by $\phi$ and $\mathbf{Y}_i^{\mathbf{M}}$ is produced upon coded aperture $\mathbf{M}$. Then, the resulting input sample is aligned as

$$\mathbf{Y}_i^{\mathbf{M}} = \mathbf{Y}_i^{\mathbf{M}} + \Phi(\phi; \mathbf{M}). \tag{9}$$

In the proposed method, the prompt network collaborates different clients with inconsistent hardware configurations. It takes effect by implicitly observing and collecting diverse coded aperture samples of all clients, and jointly learns to react to different hardware settings. The prompter regularizes the input data space and achieves the goal of coping with heterogeneity sourcing from hardware.

**Training**. As shown in Fig. 2, we demonstrate the training process of proposed FedHP by taking one global round as an example[3]. Since the prompt learning takes effect on pre-trained models, we initialize the $c$-th backbone parameters with the pre-trained model $\theta_c^p$ on local data $\mathcal{D}^{\mathbf{M}^c}$ with (7). The global prompt network $\phi_G$ is randomly initialized and distributed to the $c$-th client

$$\phi_c \leftarrow \phi_G, \ c = 1, ..., C', \tag{10}$$

where $\phi_c$ is the local prompt network, and $C'$ denotes the number of clients participated in the current global round. To enable better response of the pre-trained backbone toward the aligned input data space, we also introduce the adaptors into the transformer backbone. As shown in Fig. 2 *Step 3*, we show the architecture of the proposed adaptor, which is a *CONV-GELU-CONV* structure governed by a residual connection. We insert the adaptors behind the *LN* layers.

We perform local updates in each global round. It is composed of two stages. Firstly, we update the local prompt network $\phi_c$ for $S_p$ iterations, and fix all the other learnable parameters . The loss is

$$\ell_c = \frac{1}{N} \sum_{i=1}^{N} ||f(\theta_c^p, \epsilon_c; \mathbf{Y}_i^{\mathbf{M}^c} + \Phi(\mathbf{M}^c)) - \mathbf{X}_i||_2^2, \tag{11}$$

where we use $\epsilon_c$ to represent learnable parameters of all adaptors for $c$-th client. Secondly, we tune the adaptors for another $S_b$ iterations. Both of the pre-trained backbone and prompt network are frozen. The loss of $c$-th client shares the same formulation as (11). After the local update, FedHP uploads and aggregates the learnable parameters $\phi_c, c = 1, ..., C$ of the prompt network. Since the proposed method does not require to optimize and communicate the reconstruction backbones, the underlying cost is drastically reduced considering the marginal model size of prompt network and adpators compared with the backbone, which potentially serves as a supplied benefit of FedHP.

Compared with FedAvg, FedHP adopts the hardware prompt to explicitly align the input data representation and handle the distribution shift attributing to the coded aperture inconsistency or coded aperture distribution discrepancy.

## 3 Experiments

### 3.1 Implementation details

**Dataset**. Following existing practices [Cai et al., 2022b, Lin et al., 2022, Hu et al., 2022, Huang et al., 2021], we adopt the benchmark training dataset of CAVE [Yasuma et al., 2010], which is composed of 32 hyperspectral images with the spatial size as $512 \times 512$. Data augmentation techniques of rotation, flipping are employed, producing 205 different training scenes. For the federated learning, we equally split the training dataset according to the number of clients $C$. The local training dataset are kept and accessed confidentially across clients. Note that one specific coded aperture determines a unique dataset according to (2), the resulting data samples for each client can be much more than $205/C$. We employ the widely-used simulation testing dataset for the quantitative evaluation, which consists of ten $256 \times 256 \times 28$ hyperspectral images collected from KAIST [Choi et al., 2017]. Besides, we use the real testing data with spatial size of $660 \times 660$ collected by a SD-CASSI system [Meng et al., 2020] for the perceptual evaluation considering the real-world perturbations.

**Hardware**. We collect and will release the first Snapshot Spectral Heterogeneous Dataset (SSHD) containing a series of practical SCI systems, from three agencies, each of which offers a series of coded apertures that correspond to a unique distribution[4] as presented by federated settings in Fig. 2. No identical coded apertures exists among all systems. For the case of inconsistent mask distributions,

---

[3]We provide an algorithm of FedHP in supplementary.

[4]More illustrations and distribution visualizations of real collected coded apertures are in supplementary.

Table 1: PSNR(dB)/SSIM performance comparison. For different clients, we sample non-overlapping masks from the same mask distribution to train the model and use unseen masks randomly sampled from all clients for testing. We report $mean_{\pm std}$ among 100 trials for all methods.

| Scene | FedAvg | | FedProx | | SCAFFOLD | | FedGST | | FedHP (ours) | |
|---|---|---|---|---|---|---|---|---|---|---|
| | PSNR | SSIM | PSNR | SSIM | PSNR | SSIM | PSNR | SSIM | PSNR | SSIM |
| 1 | $31.98_{\pm0.19}$ | $0.8938_{\pm0.0025}$ | $31.85_{\pm0.21}$ | $0.8903_{\pm0.0028}$ | $31.78_{\pm0.24}$ | $0.8886_{\pm0.0025}$ | $32.02_{\pm0.14}$ | $0.8918_{\pm0.0018}$ | $\mathbf{32.31}_{\pm0.19}$ | $\mathbf{0.9026}_{\pm0.0020}$ |
| 2 | $30.49_{\pm0.21}$ | $0.8621_{\pm0.0041}$ | $29.85_{\pm0.22}$ | $0.8516_{\pm0.0037}$ | $29.81_{\pm0.19}$ | $0.8473_{\pm0.0031}$ | $30.13_{\pm0.20}$ | $0.8519_{\pm0.0038}$ | $\mathbf{30.78}_{\pm0.19}$ | $\mathbf{0.8746}_{\pm0.0034}$ |
| 3 | $31.78_{\pm0.23}$ | $0.9088_{\pm0.0019}$ | $30.80_{\pm0.23}$ | $0.8968_{\pm0.0017}$ | $30.92_{\pm0.17}$ | $0.8961_{\pm0.0014}$ | $31.19_{\pm0.22}$ | $0.8975_{\pm0.0015}$ | $\mathbf{31.62}_{\pm0.25}$ | $\mathbf{0.9109}_{\pm0.0018}$ |
| 4 | $39.39_{\pm0.23}$ | $0.9559_{\pm0.0018}$ | $39.41_{\pm0.22}$ | $0.9601_{\pm0.0013}$ | $39.32_{\pm0.20}$ | $0.9565_{\pm0.0011}$ | $38.98_{\pm0.27}$ | $0.9513_{\pm0.0020}$ | $\mathbf{39.78}_{\pm0.29}$ | $\mathbf{0.9633}_{\pm0.0017}$ |
| 5 | $28.70_{\pm0.16}$ | $0.8821_{\pm0.0044}$ | $28.14_{\pm0.16}$ | $0.8765_{\pm0.0036}$ | $28.08_{\pm0.14}$ | $0.8742_{\pm0.0032}$ | $28.53_{\pm0.16}$ | $0.8743_{\pm0.0041}$ | $\mathbf{28.92}_{\pm0.17}$ | $\mathbf{0.8935}_{\pm0.0039}$ |
| 6 | $30.53_{\pm0.30}$ | $0.9054_{\pm0.0025}$ | $30.04_{\pm0.23}$ | $0.9054_{\pm0.0024}$ | $29.87_{\pm0.21}$ | $0.9011_{\pm0.0019}$ | $30.29_{\pm0.21}$ | $0.8949_{\pm0.0022}$ | $\mathbf{30.77}_{\pm0.22}$ | $\mathbf{0.9172}_{\pm0.0019}$ |
| 7 | $30.01_{\pm0.20}$ | $0.8811_{\pm0.0027}$ | $29.60_{\pm0.20}$ | $0.8718_{\pm0.0026}$ | $29.63_{\pm0.19}$ | $0.8708_{\pm0.0027}$ | $29.89_{\pm0.18}$ | $0.8786_{\pm0.0024}$ | $\mathbf{30.44}_{\pm0.19}$ | $\mathbf{0.8884}_{\pm0.0024}$ |
| 8 | $28.60_{\pm0.31}$ | $0.8880_{\pm0.0023}$ | $27.93_{\pm0.20}$ | $0.8845_{\pm0.0018}$ | $27.74_{\pm0.31}$ | $0.8802_{\pm0.0018}$ | $28.35_{\pm0.19}$ | $0.8752_{\pm0.0016}$ | $\mathbf{28.56}_{\pm0.32}$ | $\mathbf{0.8957}_{\pm0.0021}$ |
| 9 | $31.45_{\pm0.15}$ | $0.9012_{\pm0.0019}$ | $31.29_{\pm0.15}$ | $0.8961_{\pm0.0019}$ | $31.22_{\pm0.14}$ | $0.8929_{\pm0.0014}$ | $30.80_{\pm0.12}$ | $0.8880_{\pm0.0021}$ | $\mathbf{31.34}_{\pm0.13}$ | $\mathbf{0.9043}_{\pm0.0023}$ |
| 10 | $29.04_{\pm0.13}$ | $0.8751_{\pm0.0022}$ | $28.48_{\pm0.15}$ | $0.8671_{\pm0.0035}$ | $28.59_{\pm0.13}$ | $0.8626_{\pm0.0028}$ | $28.51_{\pm0.13}$ | $0.8578_{\pm0.0024}$ | $\mathbf{29.12}_{\pm0.13}$ | $\mathbf{0.8835}_{\pm0.0021}$ |
| *Avg.* | $31.21_{\pm0.10}$ | $0.8959_{\pm0.0017}$ | $30.76_{\pm0.10}$ | $0.8900_{\pm0.0016}$ | $30.71_{\pm0.09}$ | $0.8872_{\pm0.0013}$ | $30.85_{\pm0.11}$ | $0.8858_{\pm0.0017}$ | $\mathbf{31.35}_{\pm0.10}$ | $\mathbf{0.9033}_{\pm0.0014}$ |

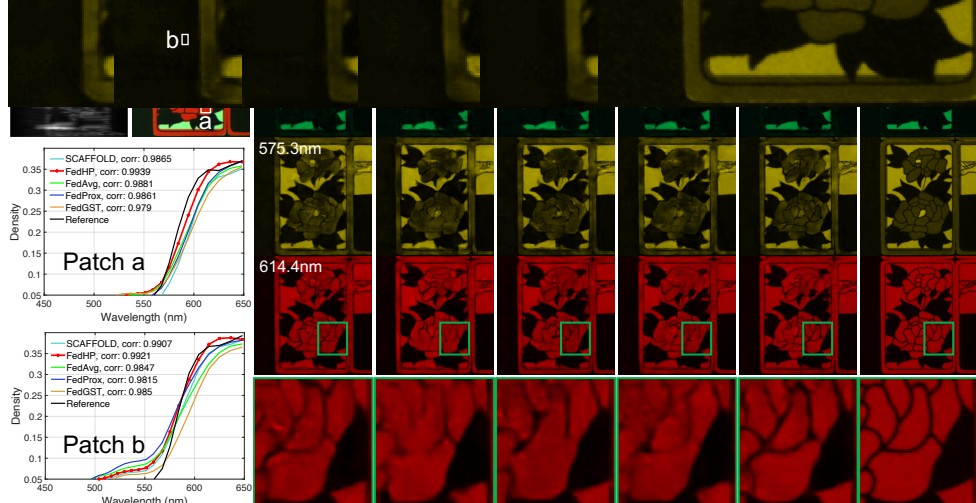

Figure 3: Reconstruction results on simulation data. The density curves compare the spectral consistency of different methods to the ground truth. We use the same coded aperture for all methods.

we directly assign hardware systems from one source to form a client. We simulate the scenario of non-overlapping masks by distributing coded apertures from one source to different clients.

**Implementation details**. We adopt MST-S [Cai et al., 2022a] as the reconstruction backbone. The prompt network is instantiated by a SwinIR [Liang et al., 2021] block. Limited by the computational resource, we set the number of clients as 3 in main comparison. We empirically find that collaborate such amount of clients can be problematic for popular federated learning methods under the very challenging scenario of data heterogeneity (see Section 3.2). For FL methods, we update all clients throughout the training, *i.e.*, $C' = C = 3$. For the proposed method, we pre-train the client backbones from scratch for $4 \times 10^4$ iterations on their local data. Notably, the total training iterations of different methods are kept as $1.25 \times 10^5$ for a fair comparison. The batch is set as 12. We set the initial learning rate for both of the prompt network and adaptor as $\alpha_p = \alpha_b = 1 \times 10^{-4}$ with step schedulers, *i.e.*, half annealing every $2 \times 10^4$ iterations. We train the model with an Adam [Kingma and Ba, 2014] optimizer ($\beta_1 = 0.9, \beta_2 = 0.999$). We use PyTorch [Paszke et al., 2017] on an NVIDIA A100 GPU.

**Compared Methods**. We compare FedHP with mainstream FL methods, including FedAvg [McMahan et al., 2017], FedProx [Li et al., 2020b], and SCAFFOLD [Karimireddy et al., 2020]. Besides, GST [Wang et al., 2022] paves the way for the robustness of the reconstruction toward multiple hardware. Thereby, we integrate this method into the FL framework, dubbed as FedGST. All methods require to train and aggregate the entire client backbones. By comparison, FedHP updates and shares the prompt network, outperforming the others with smaller amount of parameters being optimized and communicated. We adopt PSNR and SSIM [Wang et al., 2004] for the quantitative evaluation.

## 3.2 Performance

**Simulation Results**. We quantitatively compare different methods in Table 1 by considering the data heterogeneity stems from non-overlapping masks. FedHP performs better than the classic federated

Table 2: PSNR(dB)/SSIM performance comparison. Masks from each client are sampled from a *specific* distribution for training. We randomly sample non-overlapping masks (unseen to training) from all distributions for testing. We report $mean_{\pm std}$ among 100 trials for all methods.

| Scene | FedAvg | | FedProx | | SCAFFOLD | | FedGST | | FedHP (ours) | |
|---|---|---|---|---|---|---|---|---|---|---|
| | PSNR | SSIM | PSNR | SSIM | PSNR | SSIM | PSNR | SSIM | PSNR | SSIM |
| 1 | $29.15_{\pm0.09}$ | $0.8392_{\pm0.0065}$ | $23.01_{\pm0.11}$ | $0.5540_{\pm0.0069}$ | $22.99_{\pm0.13}$ | $0.5535_{\pm0.0066}$ | $29.46_{\pm0.65}$ | $0.8344_{\pm0.0067}$ | $\mathbf{30.37}_{\pm0.70}$ | $\mathbf{0.8628}_{\pm0.0084}$ |
| 2 | $28.28_{\pm0.10}$ | $0.8102_{\pm0.0052}$ | $20.91_{\pm0.08}$ | $0.4486_{\pm0.0052}$ | $20.89_{\pm0.09}$ | $0.4474_{\pm0.0055}$ | $27.89_{\pm0.36}$ | $0.7733_{\pm0.0068}$ | $\mathbf{28.67}_{\pm0.38}$ | $\mathbf{0.8160}_{\pm0.0072}$ |
| 3 | $28.42_{\pm0.11}$ | $0.8464_{\pm0.0083}$ | $17.57_{\pm0.11}$ | $0.4621_{\pm0.0082}$ | $17.58_{\pm0.12}$ | $0.4608_{\pm0.0083}$ | $28.45_{\pm0.50}$ | $0.8363_{\pm0.0073}$ | $\mathbf{29.81}_{\pm0.68}$ | $\mathbf{0.8771}_{\pm0.0066}$ |
| 4 | $36.93_{\pm0.27}$ | $0.9369_{\pm0.0036}$ | $23.08_{\pm0.25}$ | $0.4856_{\pm0.0036}$ | $23.00_{\pm0.30}$ | $0.4848_{\pm0.0038}$ | $36.12_{\pm0.50}$ | $0.9181_{\pm0.0050}$ | $\mathbf{37.37}_{\pm0.53}$ | $\mathbf{0.9395}_{\pm0.0032}$ |
| 5 | $25.84_{\pm0.07}$ | $0.8037_{\pm0.0069}$ | $18.99_{\pm0.07}$ | $0.4316_{\pm0.0082}$ | $18.99_{\pm0.06}$ | $0.4301_{\pm0.0065}$ | $26.21_{\pm0.52}$ | $0.7988_{\pm0.0081}$ | $\mathbf{27.47}_{\pm0.73}$ | $\mathbf{0.8487}_{\pm0.0011}$ |
| 6 | $27.28_{\pm0.04}$ | $\mathbf{0.8655}_{\pm0.0041}$ | $19.10_{\pm0.04}$ | $0.4077_{\pm0.0041}$ | $19.10_{\pm0.04}$ | $0.4063_{\pm0.0042}$ | $27.52_{\pm0.49}$ | $0.8384_{\pm0.0048}$ | $\mathbf{28.31}_{\pm0.45}$ | $0.8649_{\pm0.0050}$ |
| 7 | $26.81_{\pm0.09}$ | $0.8042_{\pm0.0094}$ | $20.15_{\pm0.09}$ | $0.4903_{\pm0.0093}$ | $20.14_{\pm0.09}$ | $0.4883_{\pm0.0098}$ | $26.88_{\pm0.57}$ | $0.7957_{\pm0.0073}$ | $\mathbf{28.29}_{\pm0.81}$ | $\mathbf{0.8298}_{\pm0.0108}$ |
| 8 | $25.77_{\pm0.05}$ | $\mathbf{0.8473}_{\pm0.0030}$ | $19.89_{\pm0.07}$ | $0.4402_{\pm0.0031}$ | $19.89_{\pm0.06}$ | $0.4395_{\pm0.0039}$ | $26.22_{\pm0.44}$ | $0.8206_{\pm0.0029}$ | $\mathbf{26.54}_{\pm0.45}$ | $0.8470_{\pm0.0054}$ |
| 9 | $28.30_{\pm0.09}$ | $\mathbf{0.8541}_{\pm0.0074}$ | $18.33_{\pm0.11}$ | $0.4285_{\pm0.0071}$ | $18.30_{\pm0.11}$ | $0.4269_{\pm0.0078}$ | $27.74_{\pm0.48}$ | $0.8199_{\pm0.0073}$ | $\mathbf{29.36}_{\pm0.63}$ | $0.8536_{\pm0.0054}$ |
| 10 | $26.04_{\pm0.12}$ | $0.8075_{\pm0.0035}$ | $20.06_{\pm0.12}$ | $0.3461_{\pm0.0036}$ | $20.03_{\pm0.13}$ | $0.3451_{\pm0.0036}$ | $25.72_{\pm0.22}$ | $0.7433_{\pm0.0046}$ | $\mathbf{26.78}_{\pm0.26}$ | $\mathbf{0.8111}_{\pm0.0076}$ |
| *Avg.* | $28.63_{\pm0.07}$ | $\mathbf{0.8496}_{\pm0.0041}$ | $20.85_{\pm0.07}$ | $0.5405_{\pm0.0059}$ | $20.00_{\pm0.09}$ | $0.4374_{\pm0.0040}$ | $28.24_{\pm0.39}$ | $0.8177_{\pm0.0045}$ | $\mathbf{28.98}_{\pm0.23}$ | $0.8481_{\pm0.0054}$ |

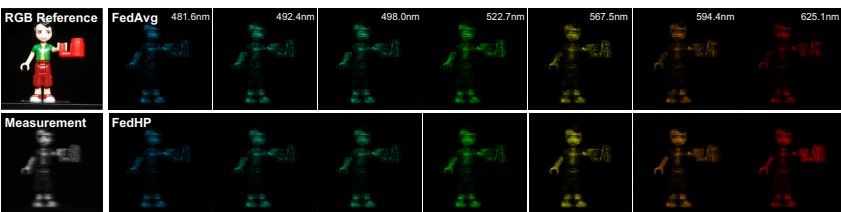

Figure 4: Visualization of reconstruction results on real data. Six representative wavelengths are selected. We use the same unseen coded aperture for both FedAvg and FedHP.

learning methods. By comparison, FedProx and SCAFFOLD only allows sub-optimal performance, which uncovers the limitations of rectifying the gradient directions in this challenging task. Besides, FedGST works inferior than FedHP, since FedGST approximates the posterior and expects coded apertures strictly follows the identical distribution, which can not be guaranteed in practice. In Fig. 3, we visualize the reconstruction results with sampled wavelengths. FedHP not only enables a more granular retrieval on unseen coded aperture, but also maintains a promising spectral consistency as shown by randomly cropped patches (*e.g.*, a, b in Fig. 3).

**Challenging Scenario of Heterogeneity**. We consider a more challenging scenario where the data heterogeneity is caused the *distinct coded aperture distributions of different clients*. We compare different methods in Table 2. All methods experience large performance degradation, among which FedProx and SCAFFOLD becomes ineffective. Intuitively, it is hard to concur the clients under the large distribution gap, while directly adjusting the input data space better tackles the problem.

**Real Results**. In Fig. 4, we visually compare the FedAvg with FedHP on the real data. Specifically, both methods are evaluated under an unseen hardware configuration, *i.e.*, coded aperture from an uncertain distribution. The proposed method introduces less distortions among different wavelengths. Such an observation endorses FedHP a great potential in collaborating hardware systems practically.

## 3.3 Model Discussion

We conduct model discussion in Table 3. Specifically, we accumulate the total cost (*e.g.*, number of parameters, GMACs, and training time) of all clients in a federated system.

**Ablation Study**. We firstly consider a scenario that trains three clients independently without FL (*FedHP w/o FL*). For a fair comparison, each client pre-trains the backbone by using the same procedure as FedHP and are then enhanced with a prompt network and adaptors for efficient fine-tuning. By comparison, FedHP enables an obvious improvement (0.6dB) by implicitly sharing the hardware and data. We then investigate the effectiveness of the prompter and adaptor to the reconstruction, respectively. By observation, directly removing the adaptor leads to limited performance descent. Using prompt network brings significant performance boost. The hardware prompter aligns the input data distributions, potentially solving the heterogeneity rooted in the input data space, considering fact that learning manifold is highly correlated with the coded apertures.

**Discussion of the client number**. In Table 4a, we discuss the power of FedHP with more real clients under the scenario of *Hardware shaking*. The performance gap between FedHP and FedAvg consistently remains with the client number increasing, which demonstrates the practicability of the FedHP for the cross-silo spectral system cooperative learning, *e.g.*, $3 \sim 5$ clients/institutions.

Table 3: Ablation study and complexity analysis under the non-overlapping masks. The PSNR (dB)/SSIM are computed among 100 testing trials. We report the model complexity and the accumulative training time of all clients (*e.g.*, $C = 3$).

| Method | Prompter | Adaptor | FL | PSNR | SSIM | #Params (M) | GMACs | Training (days) |
|---|---|---|---|---|---|---|---|---|
| FedAvg | ✗ | ✗ | ✓ | $31.21_{\pm0.10}$ | $0.8959_{\pm0.0017}$ | 0.12 | 2.85 | 10.62 |
| FedHP w/o FL | ✓ | ✓ | ✗ | $30.75_{\pm0.11}$ | $0.8890_{\pm0.0015}$ | 0.27 | 12.78 | 2.86 |
| FedHP w/o Adaptor | ✓ | ✗ | ✓ | $31.09_{\pm0.10}$ | $0.8996_{\pm0.0017}$ | 0.15 | 11.01 | 2.68 |
| FedHP w/o Prompter | ✗ | ✓ | ✓ | $19.19_{\pm0.01}$ | $0.2303_{\pm0.0008}$ | 0.12 | 2.87 | 2.54 |
| FedHP (Full model) | ✓ | ✓ | ✓ | $31.35_{\pm0.10}$ | $0.9033_{\pm0.0014}$ | 0.27 | 12.78 | 2.86 |

Table 4: Model discussions of the proposed FedHP.

(a) #Client discussion. Averaged values are reported.

| $C$ | FedAvg | | FedHP | | Performance gap | |
|---|---|---|---|---|---|---|
| 4 | 31.06 | 0.8955 | 31.33 | 0.9023 | 0.27 | 0.0068 |
| 5 | 31.05 | 0.9025 | 31.32 | 0.9029 | 0.27 | 0.0004 |

(b) Comparison with a deep Unfolding method.

| Methods | PSNR(dB) | SSIM | #Params (M) |
|---|---|---|---|
| GAP-Net | $31.07_{\pm0.20}$ | $0.8895_{\pm0.0035}$ | 3.83 |
| FedHP | $31.35_{\pm0.10}$ | $0.9033_{\pm0.0014}$ | 0.27 |

**Comparison with a deep unfolding method**. We also compare the proposed FedHP with a representative deep unfolding method of GAP-Net [Meng et al., 2023] as deep unfolding methods can be adaptable to various hardware configurations. Specifically, we use three clients and keep training and testing settings of GAP-Net the same as FedHP. As shown in Table 4b, FedHP improves by 0.28dB with only 7% model size. In fact, despite the adaptability, deep unfolding still shows limitations in solving hardware perturbation/replacement for a given system [Wang et al., 2022].

## 4 Related Work

**Hyperspectral Image Reconstruction**. In hyperspectral image reconstruction (HSI), learning deep reconstruction models [Cai et al., 2022a,b, Lin et al., 2022, Huang et al., 2021, Meng et al., 2020, Hu et al., 2022, Miao et al., 2019] has been the forefront among recent efforts due to high-fidelity reconstruction and high-efficiency. Among them, MST [Cai et al., 2022a] devises the first transformer backbone by computing spectral attention. Existing reconstruction learning strategies mainly considers the compatibility toward a single hardware instance. The learned model can be highly sensitive to the variation of hardware. To tackle this practical challenge, GST [Wang et al., 2022] paves the way by proposing a variational Bayesian learning treatment.

**Federated Learning**. Federated learning [Kairouz et al., 2021, Li et al., 2020a, Wang et al., 2021] collaborates client models without sharing the privacy-sensitive assets. However, FL learning suffers from client drift across clients attributing to the data heterogeneity issue. One mainstream [Karimireddy et al., 2020, Li et al., 2020b, Xu et al., 2021, Jhunjhunwala et al., 2023, Reddi et al., 2021] mainly focus on regularizing the global/local gradients. As another direction, personalized FL methods [Collins et al., 2021, Chen and Chao, 2022, Fallah et al., 2020, T Dinh et al., 2020, Jiang and Lin, 2023] propose to fine-tune the global model for better adaptability on clients. However, customizing the global model on client data sacrifices the underlying robustness upon data distribution shift [Wu et al., 2022, Jiang and Lin, 2023], which contradicts with our goal of emphasizing the generality across hardware and thus is not considered. In this work, we propose a federated learning framework to solve the multi-hardware cooperative learning considering the data privacy and heterogeneity, which to the best knowledge, is the first attempt of empowering spectral SCI with FL. Besides, the principle underlying this method can be potentially extended to broad computational imaging applications [Zheng et al., 2021, Liu et al., 2023a, Goudreault et al., 2023, Robidoux et al., 2021]

## 5 Conclusion

In this work, we observed an unexplored research scenario of multiple hardware cooperative learning in spectral SCI, considering two practical challenges of privacy constraint and the heterogeneity stemming from inconsistent hardware configurations. We developed a Federated Hardware-Prompt (FedHP) learning framework to solve the distribution shift across clients and empower the hardware-software co-optimization. The proposed method serves as a first attempt to exploit the power of FL in spectral SCI. Besides, we have collected a Snapshot Spectral Heterogeneous Dataset (SSHD) from multiple real spectral SCI systems. Future works may theoretically derive the convergence of FedHP and exploit the behavior of FedHP under a large number of clients. We hope this study will inspire broad explorations in this novel direction of hardware collaboration in SCI.

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

# A  Appendix / supplemental material

We provide more discussions and results of the proposed FedHP as follows

## A.1  Limitations

One of the limitations of the proposed method is the lack of the real hardares due to the privacy concern. Thus it is hard for us to perform the federated learning on a large number of the clients as in other tasks like the classification, *e.g.*, $C > 100$. This in return, motivate us to solve the practical concerns of this field. We are working on collecting more real data and will continue exploring the power of the proposed method.

## A.2  Broader Impacts

This work develops a federated learning treatment to enable the collaboration of the CASSI systems with different hardware configurations. The proposed method will practically encourage the cross-institution collaborations with emerging optical system designs engaged. By improving the robustness of the pre-trained reconstruction software backend toward optical encoders, this work will help expedite the efficient and widespread deployment of the deep models on sensors or platforms.

Table 5: Performance comparison between FedAvg and FedHP on CACTI (*e.g.*, $C = 3$).

| Methods | PSNR (dB) | SSIM |
|---------|-----------|------|
| FedAvg | $27.35_{\pm 1.22}$ | $0.9174_{\pm 0.0046}$ |
| FedHP | $27.87_{\pm 0.89}$ | $0.9192_{\pm 0.0047}$ |

## A.3  New Hardware

Our key technical contribution is to provide a new multi-hardware optimization framework adapting to hardware shift by only accessing local data. The principle underlying the proposed FedHP can be potentially extended to broad SCI applications. This work serves as a proof of concept to inspire future endeavors in a more general scope. Besides experimental results on CASSI, we also perform additional experiments by applying FedHP to another prevalent SCI system of Coded Aperture Compressive Temporal Imaging (CACTI) [Llull et al., 2013]. The results in Table 5 present a performance boost of FedHP over FedAvg baseline (under the same setting as the manuscript), demonstrating that the proposed FedHP does not particularly pertain to CASSI.

## A.4  Algorithm

The learning procedure of proposed FedHP is provided in Algorithm 1. Let us take one global round for example, the learning can be divided into four stages. (1) Initializing the global prompt network from scratch and then distributing it to local clients. Then instantiating the client backbones with the pre-trained models upon the local training dataset. The adaptors are also randomly initialized for a better adaptation of the pre-trained backbones to the aligned input data representation. (2) Local updating of the prompt network, during which all the other learnable parameters in the system are kept fixed. (3) Local updating of the adaptors. Notably, the parameters of the adaptors is only updated and maintained in local. (4) Global aggregation of the local prompt networks.

**Algorithm 1** FedHP Training Algorithm

---

**Input:** Number of global rounds $T$; Number of clients $C$; Number of client subset $C'$; Pre-trained models $\theta_c^p$, $c = 1, ..., C$; Number of local update iterations $S_p, S_b$; Random initialized parameter of prompt network $\phi_G$; Random initialized parameter of adaptors of $c$-th client $\epsilon_c$; Learning rate $\alpha_p$ of prompt network; Learning rate $\alpha_b$ of adaptors;
**Output:** $\phi_G, \epsilon_c, c = 1, ..., C$;
 1: Server Executes;
 2: Randomly choose a set of clients of number $C'$;
 3: **for** $t = 1, ..., T$ **do**
 4:     **for** $c \in C'$ in parallel **do**
 5:         Send global prompt network $\phi_G$ to $\phi_c$;
 6:         $\phi_c \leftarrow \text{LocalTraining}(\theta_c^p, \epsilon_c, \phi_c)$;
 7:     **end for**
 8:     $\phi_G \leftarrow \sum_{c=1}^{c=C'} \frac{|\mathcal{D}_c|}{|\mathcal{D}|} \phi_c$;
 9: **end for**
10: **return** $\phi_G$;
11: LocalTraining$(\theta_c^p, \epsilon_c, \phi_c)$;
12: **for** $s = 1, ..., S_p$ **do**
13:     $\phi_c \leftarrow \phi_c - \alpha_p \nabla \ell(\theta_c^p, \epsilon_c, \phi_c)$ using $\ell_c = \frac{1}{N} \sum_{i=1}^{N} ||f(\theta_c^p, \epsilon_c; \mathbf{Y}_i^{\mathbf{M}^c} + \Phi(\mathbf{M}^c)) - \mathbf{X}_i||_2^2$;
14: **end for**
15: **for** $s = 1, ..., S_b$ **do**
16:     $\epsilon_c \leftarrow \epsilon_c - \alpha_b \nabla \ell(\theta_c^p, \epsilon_c, \phi_c)$ using $\ell_c = \frac{1}{N} \sum_{i=1}^{N} ||f(\theta_c^p, \epsilon_c; \mathbf{Y}_i^{\mathbf{M}^c} + \Phi(\mathbf{M}^c)) - \mathbf{X}_i||_2^2$;
17: **end for**
18: **return** $\phi_c$ to server;

---

### A.5 Visualizations

In this section, we provide more visualization results of different methods. In Figs. 5∼6, we present the reconstruction results of different methods under the scenario of `hardware shaking`, *i.e.*, the data heterogeneity is naively induced from the different CASSI instances across clients. FedHP enables more fine-grained details retrieval. Besides, we compare the spectral density curves on selected representative spatial regions. The higher correlation to the reference, the better spetrum consistency with the ground truth. In Figs. 7∼9, we show additional real reconstruction results of FedAvg and FedHP on selected wavelengths. By comparison, FedAvg fails to reconstruct some content, while the proposed FedHP allows a more granular result.

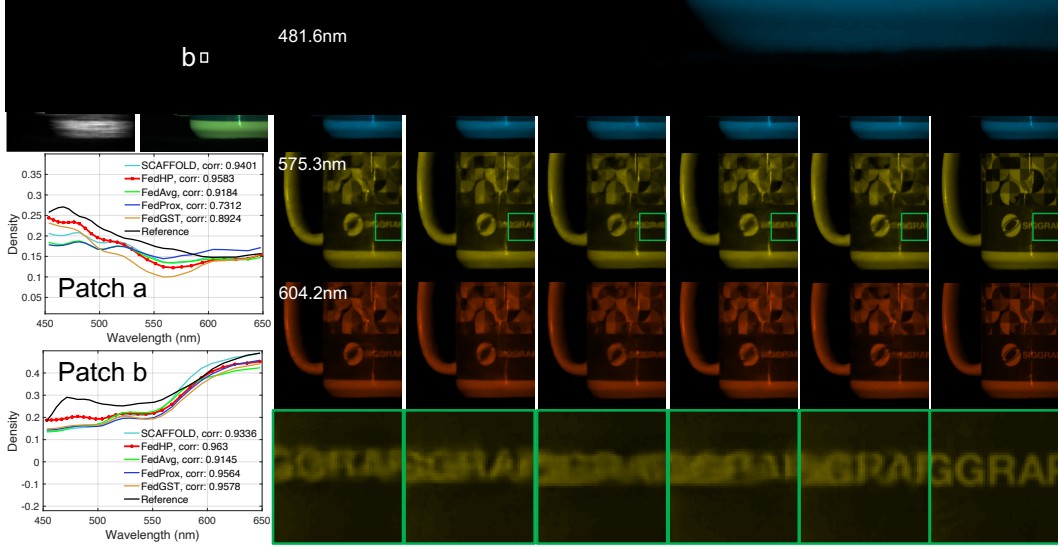

Figure 5: Reconstruction results on simulation data. The density curves compares the spectral consistency of different methods to the ground truth. We use the same coded aperture for all methods.

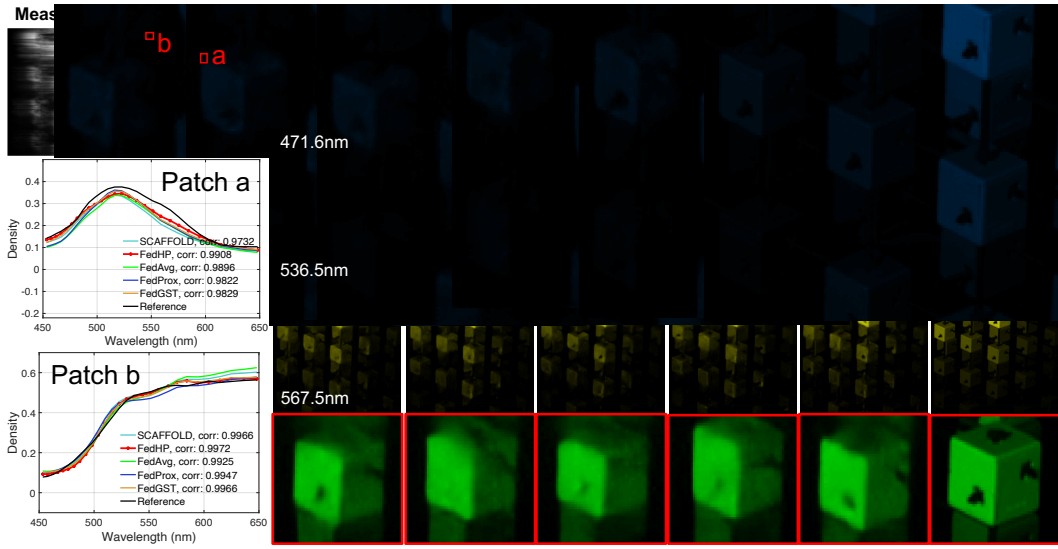

Figure 6: Reconstruction results on simulation data. The density curves compares the spectral consistency of different methods to the ground truth. We use the same coded aperture for all methods.

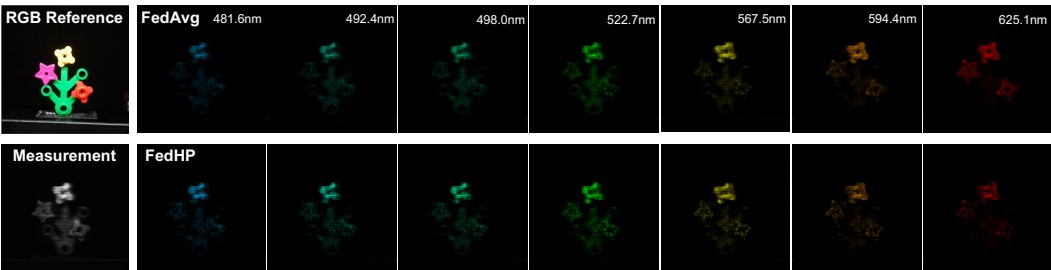

Figure 7: Visualization of reconstruction results on real data. Seven (out of 28) representative wavelengths are selected. We use the same unseen coded aperture for both FedAvg and FedHP.

In Figs. 8, we visualize the different distributions of coded apertures in distinct clients under the scenario of the distribution shift of coded apertures among different clients leads to the data heterogeneity among different local input dataset. This mimics a very challenging scenario where in different clients (*e.g.*, research institutions), the corresponding CASSI systems source from different manufacturers. The proposed FedHP allows a potential collaboration among different institutions for the hyperspectral data acquisition for the first time despite the large distribution gap. By comparison, classic methods of FedProx [Li et al., 2020b] or SCAFFOLD [Karimireddy et al., 2020] fail to provide reasonable retrieval results.

## A.6 Data Privacy Protection

FedHP inherently addresses privacy from different perspectives. (1) Hardware decentralization: In the FedHP framework, real hardware configurations (*e.g.*, real masks) remain confidential to the local clients. This design makes it difficult to reverse-engineer the pattern or values of the real mask without direct sharing. (2) Raw data decentralization: FedHP maintains a private hyperspectral dataset for each client. The hyperspectral images are processed locally (*e.g.*, encoding or data augmentation) and never leaves the client, thereby minimizing the risk of exposure. (3) Training process decentralization: FedHP only collects the local updates from the prompt network, which are then shared with the central server. The local updates are anonymized and aggregated without accessing underlying data, preventing any tracing back to the data source and thus protecting confidentiality. In Table 3, we quantitatively compared the performance of the proposed "FedHP" and "FedHP w/o FL" under privacy-constrained environments. FedHP demonstrates a dB average improvement, showcasing its robust model performance and offering a significant privacy advantage that aligns with regulations restricting data sharing.

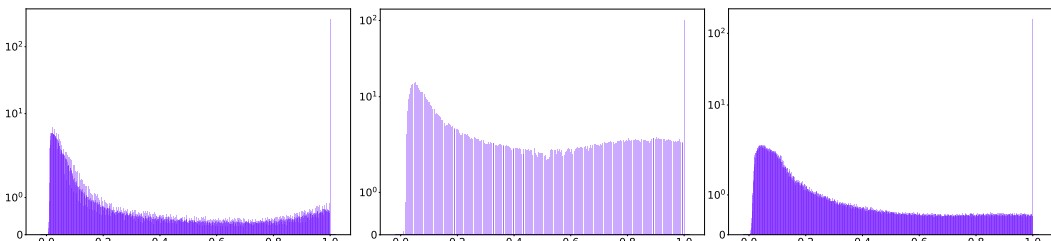

Figure 8: Coded aperture distributions across Clients $1 \sim 3$ under the scenario of `manufacturing discrepancy`. The symmetrical logarithm scale is employed for better visualization.

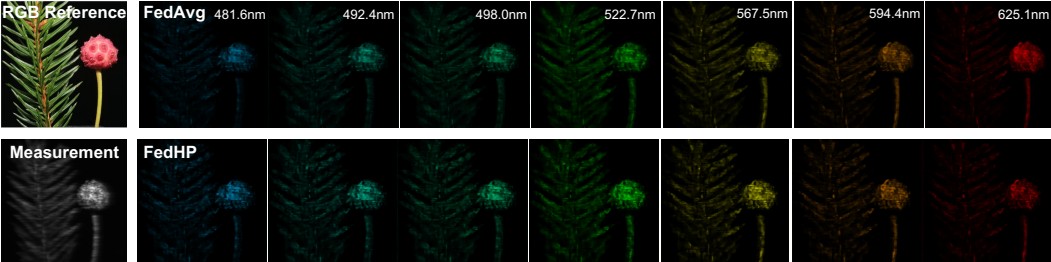

Figure 9: Visualization of reconstruction results on real data. Seven (out of 28) representative wavelengths are selected. We use the same unseen coded aperture for both FedAvg and FedHP.

### A.7 Statistical Analysis

We further conducted a statistical analysis using a paired t-test to compare the PSNR and SSIM values from FedHP and FedAvg. We define the hypotheses as follows: (1) Null hypothesis ($H_0$): there is no significant difference in the PSNR and SSIM values between FedAvg and proposed FedHP. (2) Alternative hypothesis ($H_a$): there is a significant difference in the PSNR and SSIM values between FedAvg and proposed FedHP.

We calculated the differences based on the averaged PSNR and SSIM values for each scene from both FedAvg and FedHP, resulting in ten differences values for PSNR ($d_{\text{PSNR}}$) and SSIM ($d_{\text{SSIM}}$). We performed the paired t-test using $t = \frac{\bar{d}}{s_d/\sqrt{n}}$, where $\bar{d}$ denotes the mean of the difference values for either PSNR or SSIM, $s_d$ is the standard deviations, and $n$ is the number of the paired observations.

We calculated the p-value upon the t-distribution for a two-tailed test using the formula `p-value` $= 2 \times P(T > |t|)$, where $P(T > |t|)$ denotes the probability that a t-distributed random variable with $n - 1$ degrees of freedom exceeds the absolute value of the observed t-statistic.

For PSNR, we observe $t = 2.50$ and p-value is $0.034$. Since the p-value is less than the typical significance level of $0.05$. Therefore, we reject the null hypothesis ($H_0$) and conclude that there is a statistically significant difference between the PSNR values of FedAvg and FedHP. For SSIM, we observe $t = 7.39$ and p-value is $0.00004$. The p-value of is significantly less than $0.05$, indicating a very strong statistically significant difference between the SSIM values of FedAvg and FedHP. The test results in PSNR and SSIM confirms that the performance gap between FedHP and FedAvg is statistically significant.

