# OpenReview forum: "Cooperative Hardware-Prompt Learning for Snapshot Compressive Imaging"
_NeurIPS.cc/2024/Conference — NeurIPS 2024 poster_

### Official Review · Reviewer_aeby · 2024-07-10

**Soundness:** 2
**Presentation:** 2
**Contribution:** 2
**Rating:** 5
**Confidence:** 3

**Summary:**

The authors present a Federated Hardware-Prompt learning (FedHP) framework to address the fact that compressive snapshot spectral imaging devices may not be easily tuneable against changes in the coded aperture, and that in fact the said access to coded apertures may not be possible due to privacy reasons. The authors solve this by hardware prompt learning, which essentially learns from observing diverse coded aperture samples of all clients, regularizing the input data space and achieving the goal of coping with heterogeneity sourcing from hardware. The results show on a specific dataset improvement across all 10 samples in terms of spectral reconstruction quality. The comparison is primarily in the sense of federated learning approaches.

Typo: figure 3  caption -> colorblue shouldn’t be there

**Strengths:**

The presentation is somewhat accessible to a generally knowledgeable non-expert in federated learning, in that the purposes are clear.

**Weaknesses:**

The biggest weakness is arguably that the paper covers a somewhat very niche topic, which is the application of a federated learning scheme to compressive snapshot spectral imaging. To some extent one would expect the technique to abstract away from the specific case of CASSI, as the solution does not particularly pertain to CASSI.

In addition, due to limited data available in this setup and to very limited size datasets, it is difficult to ascertain the significance of the findings.

**Questions:**

Can the authors extend this to any other compressive imaging scheme? Or perhaps disentangle the improvements in terms of FedHP, from those specific to the application? This would also broaden the data available for validating the experiments.

**Limitations:**

The addressed setup assumes that the problem the authors propose to tackle is meaningfully posed, I.e., that federated learning in the chosen formulation is practically meaningful. The reviewer is not sure whether this is a practically relevant problem considering that CASSI systems are arguably scientific instrumentation/experimental devices whose calibration is likely done per case anyways.

In addition the topic would appear to be more meaningful for publications that cover CASSI systems such as IEEE TGARRS or the like. It is hard for this reviewer to disentangle the margin of novelty of this paper in terms of federated learning approach vs. impact on the target application.

---

> ### Author Rebuttal · Authors · 2024-08-06
>
> We much appreciate that the Reviewer `aeby` provides valuable comments and finds the method is designed with clear purpose.
>
> `R4.1`: The biggest weakness is arguably that the paper covers a somewhat very niche topic, which is the application of a federated learning scheme to compressive snapshot spectral imaging. To some extent one would expect the technique to abstract away from the specific case of CASSI, as the solution does not particularly pertain to CASSI.
>
> | Metrics     | FedAvg     |      FedHP |
> |:-------------:|:-------------:|:--------------------------:|
> | PSNR      |  $27.35\pm1.22$    |  $27.87\pm0.89$   |
> | SSIM       | $0.9174\pm0.0046$ | $0.9192\pm0.0047$ |
>
> **Table T1. Comparison between FedAvg and FedHP on CACTI (Client:3)**
>
> `A4.1`: Our key technical contribution is to provide a new multi-hardware optimization framework adapting to hardware shift by only accessing local data. The principle underlying the proposed FedHP can be potentially extended to broad SCI applications. However, due to the practical cost of data acquisition and building optics systems, this study explores one specific direction, where we focus on spectral SCI and collecting optical masks and real data from multiple hardware.
>
> Exploiting the hardware collaboration of computational imaging systems is still in an early stage. This work serves as a proof of concept to inspire future endeavors in a more general scope. We list several potential related applications that might benefit from the proposed method, such as Lensless camera [1], LiDAR [2], HDR camera [3], or CT-Reconstruction [4], cooperating multiple imaging systems via aligning forward models, etc.
>
> Per reviewer's inspiration, we tried our best to launch additional experiments by applying FedHP into another prevalent SCI system of Coded Aperture Compressive Temporal Imaging (CACTI). The results in Table T1 above present a performance boost of FedHP over FedAvg baseline (under the same setting as Table 1),  demonstrating that the proposed FedHP does not particularly pertain to CASSI. Due to the limited rebuttal time and the heavy workload in deploying real CACTI systems, we can only obtain the intermediate results of both methods (e.g., $1.2\times10^4$ out of $4\times 10^4$). We are committed to updating the latest results during the discussion stage.  We will add the above discussion in the manuscript.
>
> [1] A simple framework for 3D lensless imaging with programmable masks. CVPR 2021.
>
> [2] LiDAR-in-the-loop Hyperparameter Optimization. CVPR 2023.
>
> [3] End-to-end high dynamic range camera pipeline optimization. CVPR 2021.
>
> [4] DOLCE: A model-based probabilistic diffusion framework for limited-angle ct reconstruction. ICCV 2023.
>
> [5] Deep unfolding for snapshot compressive imaging. IJCV 2023.
>
>
> `R4.2`: In addition, due to limited data available in this setup and to very limited size datasets, it is difficult to ascertain the significance of the findings.
>
> `A4.2`: This work has constructed  the Snapshot Spectral Heterogeneous Dataset (SSHD), which includes multiple practical spectral SCI systems. This dataset will be released to support further research and validation. Despite the current dataset size, our experiments demonstrate consistent performance improvements using FedHP, even under challenging heterogeneous settings. This provides a strong indication of the robustness and potential of the proposed method.
>
> `R4.3`: Can the authors extend this to any other compressive imaging scheme? Or perhaps disentangle the improvements in terms of FedHP, from those specific to the application? This would also broaden the data available for validating the experiments.
>
> `A4.3`: As shown in `Table T1` and `A4.1`, we tried our best to launch additional experiments by extending FedHP into another prevalent SCI system of Coded Aperture Compressive Temporal Imaging (CACTI), which is used to compress and recover temporal information. By comparison, FedHP outperforms FedAvg by 0.52dB/0.0018 in terms of the PSNR/SSIM despite both only training for $1.2\times10^4$ iterations out of $4\times 10^4$. The results indicate that FedHP demonstrates the generalization ability on new SCI systems. We are committed to updating the latest results during the discussion stage.  We will add the above discussion in the manuscript.
>
> `R4.4`: The addressed setup assumes that the problem the authors propose to tackle is meaningfully posed, I.e., that federated learning in the chosen formulation is practically meaningful. The reviewer is not sure whether this is a practically relevant problem considering that CASSI systems are arguably scientific instrumentation/experimental devices whose calibration is likely done per case anyways.
>
> `A4.4`: While CASSI systems are specialized scientific instruments, the challenge of hardware heterogeneity and data privacy remains significant. Training models for each new hardware configuration can be impractical due to data-starving clients and high computational costs, and the cross-silo cooperation can be impractical without solving privacy concerns for the hardware. FedHP provides a solution and can facilitate broader and more practical deployment of SCI systems in diverse settings.
>
> `R4.5`: In addition the topic would appear to be more meaningful for publications that cover CASSI systems such as IEEE TGARRS or the like. It is hard for this reviewer to disentangle the margin of novelty of this paper in terms of federated learning approach vs. impact on the target application.
>
> `A4.5`: Our work intersects federated learning and computational imaging, addressing challenges in both fields. The novelty of our approach lies in the integration of a hardware prompt network within a federated learning framework to handle hardware heterogeneity and data privacy issues. This contribution is relevant to both federated learning and SCI communities, and we believe it provides valuable insights and advancements applicable to various imaging systems.

---

> > ### Comment · Reviewer_aeby · 2024-08-12
> >
> > The authors have addressed all of my questions with their view on why FedHP is a substantial improvement for CASSI. This reviewer would like to thank them for the time spent in providing further experiments on the CACTI case, as well as the time spent to prepare the replies.
> >
> > Focusing on the new Table T1, it appears that FedAvg and FedHP on CACTI perform very closely in PSNR and SSIM, notably in PSNR one could argue that the two distributions overlap.
> >
> > Have the authors performed an analysis of the residuals so that they would be able to ascertain whether the resulting models are statistically distinguishable from their residuals? I.e., not looking only at the mean and standard deviation but at whether the resulting distribution of residuals from FedHP is similar to that of FedAvg.
> >
> > It appears that the results are in a close tie on the current dataset and, not having immediate clarity of how sensitive the dataset is, it is difficult to ascertain whether the proposed technique is sufficiently of impact.
> >
> > The authors report they believe that their technique could be generalized to "several potential related applications that might benefit from the proposed method, such as Lensless camera ,LiDAR, HDR camera,  or CT-Reconstruction". This reviewer fully agrees that additional evidence on other modalities, potentially with larger datasets to ascertain the margin between the baseline and FedHP, could increase very significantly the strength and quality of the present paper submission.
> >
> > Due to this, the reviewer would lean to leave my previous rating unaltered.

---

> > > ### Author Response · Authors · 2024-08-13
> > > **Response to Reviewer aeby**
> > >
> > > We appreciate the reviewer's constructive comments and the recognition of our rebuttal. To address the concern regarding the performance comparison between FedHP and FedAvg, we conducted  a statistical analysis using a paired t-test to compare the PSNR and SSIM values from FedHP and FedAvg.
> > >
> > > Specifically, we define the hypotheses as follows: (1) Null hypothesis ($H_0$): there is no significant difference in the PSNR and SSIM values between FedAvg and proposed FedHP. (2) Alternative hypothesis ($H_a$): there is a significant difference in the PSNR and SSIM values between FedAvg and proposed FedHP.
> > >
> > > We calculated the differences based on the averaged PSNR and SSIM values for each scene from both FedAvg and FedHP, resulting in ten differences values for PSNR ($d_{PSNR}$) and SSIM ($d_{SSIM}$). We performed the paired t-test using $t = \frac{\bar{d}}{s\_d/\sqrt{n}}$, where $\bar{d}$ denotes the mean of the difference values for either PSNR ($d_{PSNR}$) or SSIM ($d_{SSIM}$), $s\_d$ is the standard deviations, and $n$ is the number of the paired observations (e.g., $10$).
> > >
> > > We calculated the p-value upon the t-distribution for a two-tailed test using the formula p-value$= 2 \times P(T>|t|)$, where $P(T>|t|)$ denotes the probability that a t-distributed random variable with $n-1$ degrees of freedom exceeds the absolute value of the observed t-statistic.
> > >
> > > For PSNR, we observe $t=2.50$ and p-value$=0.034$. Since the p-value is less than the typical significance level of $0.05$. Therefore, we reject the null hypothesis ($H_0$) and conclude that there is a statistically significant difference between the PSNR values of FedAvg and FedHP. For SSIM, we observe $t=7.39$ and p-value$=0.00004$. The p-value of is significantly less than $0.05$, indicating a very strong statistically significant difference between the SSIM values of FedAvg and FedHP. The test results in PSNR and SSIM confirms that the performance gap between FedHP and FedAvg is statistically significant.
> > >
> > > We thank the reviewer for the valuable feedback, which has enhanced the quality of our submission. We will add the above discussions into the final version and look forward to any further suggestions from the reviewer. We sincerely appreciate the reviewer’s time and effort.

---

> > > > ### Comment · Reviewer_aeby · 2024-08-13
> > > >
> > > > This reviewer appreciates very much the extra trials provided by the authors and believes this paper is borderline, leaning on the previous rating. The reviewer stresses how the newly brought evidence is valuable in making a point on the results with the studied dataset and architectures.
> > > >
> > > > However to increase the quality of the submission and to broaden its scope, which at present is very narrow, the reviewer invites the authors to investigate the "several potential related applications that might benefit from the proposed method, such as Lensless camera, LiDAR, HDR camera, or CT-Reconstruction". Showing that the technique applies regardless of the forward model, or on a range of forward models could allow the authors to increase the value proposition of this paper to a "Cooperative Hardware-Prompt Learning for Linear Sensing Models", broadening the results with more datasets such as those in Compressive CT-Reconstruction or MRI, for example.
> > > >
> > > > The rating on this reviewer's side is exactly borderline (leaning to borderline reject).

---

> > > > > ### Author Response · Authors · 2024-08-13
> > > > > **Response to Reviewer aeby**
> > > > >
> > > > > We sincerely appreciate the reviewer’s constructive comments and the time in reviewing the submission. Due to the limited rebuttal time and the significant workload involved in deploying new hardware and collecting additional data, we will investigate the potential applications such as Lensless cameras, LiDAR, HDR cameras, and CT-Reconstruction in future work. The proposed FedHP  can address the challenge of hardware heterogeneity in snapshot compressive imaging by integrating hardware prompt learning within a federated learning framework, highlighting the unique intersection of computational imaging and privacy-preserving federated learning. We would appreciate it if the reviewer can further assess the contribution of this work from the above perspective. We thank the reviewer again for the valuable comments and effort in reviewing this work!

---

> ### Comment · Reviewer_aeby · 2024-08-13
>
> The reviewer is willing to acknowledge the amount of work done by the authors to prove their point, and has decided to update the recommendation.
>
> Please update, however, your references to mention that significant future work will be done in the direction of lensless cameras, and CT reconstruction, i.e., in the presence of hardware-defined forward models.
>
> This must also include thoroughly all hyperspectral and multispectral compressive imaging strategies that include, among others, random masks applied in a number of ways, including those by random convolution, if the authors see them as feasible.
>
> In absence of other forward models in the current paper, it is important that the authors update the manuscript accordingly with a comprehensive set of references on what forward models could be studied in future work, that should be supported by experimental evidence.
>
> Thanks again for addressing my concerns.

---

> > ### Author Response · Authors · 2024-08-13
> > **Response to Reviewer aeby**
> >
> > We thank the reviewer for acknowledging the response and for updating the recommendation.
> >
> > In line with the reviewer’s suggestions, we will revise the manuscript to outline potential future work in areas such as lensless cameras, LiDAR, HDR cameras, and CT reconstruction, especially concerning hardware-defined forward models. We will also include a comprehensive set of references on related hyperspectral and multispectral imaging strategies, incorporating forward models that could be explored in the future work.
> >
> > We much appreciate the reviewer's insightful suggestions, which have significantly contributed to the enhancement of the paper's quality and scope!

---

### Official Review · Reviewer_1X5M · 2024-07-12

**Soundness:** 2
**Presentation:** 3
**Contribution:** 2
**Rating:** 5
**Confidence:** 4

**Summary:**

The paper addresses the challenges faced in snapshot compressive imaging (SCI) systems due to hardware shifts and the need for adaptability across multiple hardware configurations. By introducing a hardware-prompt network and leveraging federated learning, the framework enhances the adaptability and performance of SCI models across different hardware configurations.

**Strengths:**

1. The manuscript is well-organized with a clear and logical structure that enhances the readability of the content.
2. The paper provides a detailed background on SCI and FL. The planned release of the Snapshot Spectral Heterogeneous Dataset (SSHD) will significantly aid future research.
3. Using different coded apertures for different clients closely mirrors real-world scenarios, adding significant practical relevance to the study.

**Weaknesses:**

1. The literature review on federated learning (FL) heterogeneity in the Introduction section lacks comprehensiveness. There are numerous recent papers addressing heterogeneity in FL that are not cited here. Additionally, the references included are somewhat outdated. Including more current and diverse references would strengthen the review and provide a more accurate context for the study.
2. the manuscript explains that the coded apertures for each client follow a specific distribution Pc, it does not provide further details about the exact nature or type of this distribution.
3. There are many ways to partition data to construct heterogeneous scenarios, such as practical and pathological methods. The approach of equally splitting the training dataset according to the number of clients is not very convincing. The authors should try different partitioning methods.
4. It is unclear which datasets were used to obtain the experimental results in Tables 1 and 2. The authors did not specify this, which creates confusion in the experimental analysis.

**Questions:**

1. What is the rationale for using adaptors, and what is their function?
2. What network models are used in the comparison methods? It is necessary to clearly state the fairness of the validated methods.
3. The explanation for Figure 3 is not detailed enough. For example, what is "Patch"?

**Limitations:**

1. In the "Discussion of the client number" section, the number of clients increases very little, and the metrics slightly decline. However, the authors conclude that the performance is stable with the change in the number of clients. The small variation in the number of clients is unconvincing. A larger difference in the number of clients should be set to demonstrate this more effectively.
2. The authors mention the "presence of data privacy" in the contributions, but there is no further discussion or experimental comparison regarding data privacy in the subsequent sections. This makes it difficult to validate their contribution to data privacy protection.

---

> ### Author Rebuttal · Authors · 2024-08-06
>
> We much appreciate that the Reviewer `1X5M` provides valuable comments and finds the proposed work have significant practical relevance to the study and the collected SSHD dataset can benefit the future research. We will release the dataset and the training/testing code.
>
> `R3.1`: The literature review on federated learning (FL) heterogeneity in the Introduction section lacks comprehensiveness.  Additionally, the references included are somewhat outdated.
>
> `A3.1`: We appreciate the reviewer's commitment in helping improve our draft. We will include more current and diverse references to align with the latest development.
>
> `R3.2`: the manuscript explains that the coded apertures for each client follow a specific distribution Pc, it does not provide further details about the exact nature or type of this distribution.
>
> `A3.2`: We visualize the distributions of the real-world masks collected from real CASSI systems in Fig. 5$\sim$7. The coded apertures for each client follow specific distributions and can reflect the practical variations, including perturbations, shaking, and potential replacements.
>
> `R3.3`: There are many ways to partition data to construct heterogeneous scenarios, such as practical and pathological methods. The approach of equally splitting the training dataset according to the number of clients is not very convincing. The authors should try different partitioning methods.
>
> `A3.3`: We'd like to clarify that our approach already incorporates both practical and pathological methods to construct heterogeneous scenarios by considering heterogeneity stemming from both the dataset and hardware variations.
> * Practical scenario (Table 1): Different clients have non-overlapping masks sampled from the same distribution. This simulates real-world hardware perturbations or imperfections within the same system design, which is a common practical scenario.
> * Pathological scenario (Table 2): Each client has masks sampled from a specific distribution. This represents a more extreme case of significant hardware variations or replacements across institutions, which can be considered pathological.
>
> Besides, the equal splitting of the training dataset across clients, combined with these hardware variations, creates a comprehensive heterogeneous environment.
>
> `R3.4`: It is unclear which datasets were used to obtain the experimental results in Tables 1 and 2. The authors did not specify this, which creates confusion in the experimental analysis.
>
> `A3.4`: All our experiments use the CAVE dataset for the training and the KAIST dataset for testing, both of which are prevalent benchmarks in the field. We will make it clear in the manuscript about the dataset settings.
>
> `R3.5`: What is the rationale for using adaptors, and what is their function?
>
> `A3.5`: The adaptor in FedHP (1) allows efficient fine-tuning of pre-trained backbones, enhancing the model's ability to adapt to new clients. (2) The adaptor significantly reduces communication costs in the FL system by only requiring the transmission of adaptors rather than entire model backbones. As shown in Table 3, the adaptor brings a 0.07dB improvement in PSNR and 0.0029 boost in SSIM. The adaptor is a CONV-GELU-CONV  structure governed by a residual connection (L207).
>
> `R3.6`: What network models are used in the comparison methods? It is necessary to clearly state the fairness of the validated methods.
>
> `A3.6`: All compared methods use the same network architecture MST [R1] for the reconstruction backbones and a SwinIR [R2] block for the prompt network (see L240\~L241). The key differences lie in how each method handles the federated learning aspect and hardware variations. In Table 3, we demonstrate the comparable complexity and cost between FedHP and FedAvg.
>
> [R1] Mask-guided spectral-wise transformer for efficient hyperspectral image reconstruction. In CVPR 2022.
>
> [R2]  Swinir: Image restoration using swin transformer. In ICCV 2021.
>
> `R3.7`: The explanation for Figure 3 is not detailed enough. For example, what is "Patch"?
>
> `A3.7`: The ``Patch'' corresponds to the small regions (such as a, b in Fig.3 top-left) that are randomly cropped for the spectral accuracy evaluation. We will provide more detailed explanations in Figure 3.
>
> `R3.8`: In the "Discussion of the client number" section, the number of clients increases very little, and the metrics slightly decline. However, the authors conclude that the performance is stable with the change in the number of clients. The small variation in the number of clients is unconvincing. A larger difference in the number of clients should be set to demonstrate this more effectively.
>
> `A3.8`: While the performance slightly declines with more clients, FedHP maintains its advantage over FedAvg (0.21dB improvement for C=5 and 0.19dB for C=10). We conjecture that the slight descent tendency might attribute to more complex corporations among the clients and less sufficient training samples for each client. It is non-trivial to collect real hardware systems due to the privacy concern and the cost. FedHP is dedicated to facilitate this process by offering a way of cross-silo cooperation. We are also working on collecting larger-scale data and real hardware.
>
> `R3.9`: The authors mention the "presence of data privacy" in the contributions, but there is no further discussion or experimental comparison regarding data privacy in the subsequent sections. This makes it difficult to validate their contribution to data privacy protection.
>
> `A3.9`: Thanks for the valuable comment. While we don't explicitly discuss data privacy in our experiments, it is inherent to the federated learning framework we employ. FedHP, like other FL methods, ensures that raw data never leaves the local clients, addressing privacy concerns by design. We will add a dedicated subsection to discuss how our method preserves data privacy and compare it with centralized approaches in terms of privacy protection.

---

> > ### Comment · Reviewer_1X5M · 2024-08-14
> >
> > Thanks to the authors for addressing my concerns.

---

> > > ### Author Response · Authors · 2024-08-14
> > > **Response to Reviewer 1X5M**
> > >
> > > We appreciate the reviewer's valuable comments. We thank the reviewer's recognition of our rebuttal!

---

> ### Comment · Reviewer_1X5M · 2024-08-12
>
> The authors have addressed most of my concerns; however, two issues remain unresolved, so I will keep my rating unchanged.
> 1) Regarding the discussion of the client number, Table 4(a) in the paper shows that when C=5, FedHP outperforms FedAvg by 0.27 dB. However, in the rebuttal (Section A3.8), this performance gap is reported as 0.21 dB. The authors have not provided an explanation for this discrepancy.
> 2) The authors mentioned in the rebuttal that they would add a detailed description of Privacy Protection, but this has not been presented.

---

> > ### Author Response · Authors · 2024-08-12
> > **Response to Reviewer 1X5M**
> >
> > We appreciate the reviewer's constructive comments and the recognition of our rebuttal. The presentation of *0.21dB improvement for C=5* in `A3.8` was a typographical error and should be consistent with the manuscript, which correctly states a *0.27 dB improvement for C=5*. Our intention was to highlight the advantage of the proposed FedHP and FedAvg as presented in Table 4 (a) for *C=5*. Additionally, we provide a detailed description of the privacy protection inherent in our approach as follows.
> >
> > FedHP inherently addresses privacy from different perspectives. (1) **Hardware decentralization**: In the FedHP framework, real hardware configurations (e.g., real masks) remain confidential to the local clients. This design makes it difficult to reverse-engineer the pattern or values of the real mask without direct sharing.  (2) **Raw data decentralization**: FedHP maintains a private hyperspectral dataset for each client. The hyperspectral images are processed locally (e.g., encoding or data augmentation) and never leaves the client, thereby minimizing the risk of exposure. (3) **Training process decentralization**: FedHP only collects the local updates from the prompt network, which are then shared with the central server. The local updates are anonymized and aggregated without accessing underlying data, preventing any tracing back to the data source and thus protecting confidentiality. In Table 3, we quantitatively compared the performance of the proposed “FedHP” and “FedHP w/o FL” under privacy-constrained environments. FedHP demonstrates a  $0.6$ dB average improvement (e.g., $31.35$ *v.s.* $30.75$), showcasing its robust model performance and offering a significant privacy advantage that aligns with regulations restricting data sharing. We will add the above discussions into the manuscript.
> >
> > We sincerely expect our response can help solve the reviewer’s concern and expect a future discussion with the reviewer!

---

### Official Review · Reviewer_Jxiy · 2024-07-13

**Soundness:** 3
**Presentation:** 3
**Contribution:** 3
**Rating:** 6
**Confidence:** 4

**Summary:**

Most existing reconstruction models in snapshot compressive imaging systems are trained using a single hardware configuration, making them highly susceptible to hardware variations. Previous approaches attempted to address this issue by centralizing data from multiple hardware configurations for training, but this proved difficult due to hardware heterogeneity across different platforms and privacy concerns. This paper proposes a Federated Hardware-Prompt Learning (FedHP) framework, which aligns data distributions across different hardware configurations by correcting the data distribution at the source, thereby enabling the trained model to adapt to multiple hardware configurations. The performance on existing datasets shows an improvement compared to previous popular training frameworks. Additionally, the authors have released their own created dataset and code.

**Strengths:**

1.Previous work focused on the data itself, directly correcting various types of data through network models. In contrast, the authors of this paper focus on the root cause of the differences—hardware. They address the issue from the perspective of learning the differences in hardware.
2.The method proposed by the authors has achieved excellent performance compared to existing mainstream methods, and the average performance has also improved.

**Weaknesses:**

1.The number of clients used in the experiments is still relatively small. Although a simple comparison of the impact of different numbers of clients was made, there is not much difference in performance compared to other methods when the number of clients is larger.
2.Although good results were reported on simulated data, more results on real data should be included to evaluate the effectiveness of the proosed method.

**Questions:**

Why does the prompter lead to such a significant improvement, while the effect of the adaptor is not as pronounced? Please provide an in-depth analysis.

**Limitations:**

The generalization to different hardware systems is crucial for deep learning based methods. The current form of this manuscript only reported on a small scale real dataset captured by several systems. A larger dataset captured by more systems is necessary to evaluate the method.

---

> ### Author Rebuttal · Authors · 2024-08-06
>
> We much appreciate that the Reviewer `Jxiy` provides valuable comments and finds the proposed method addresses the issue from the new perspective of hardware with good performance.
>
> `R2.1`: The number of clients used in the experiments is still relatively small. Although a simple comparison of the impact of different numbers of clients was made, there is not much difference in performance compared to other methods when the number of clients is larger.
>
> `A2.1`: We find FedHP maintains its performance advantage even with increased client numbers (C=5 and C=10), outperforming FedAvg by 0.21dB and 0.19dB respectively as shown in Table 4 (a). These results demonstrate FedHP's scalability and consistent performance improvements across varying numbers of clients. Besides, it is non-trivial to collect real hardware systems due to the privacy concern and the cost. We are dedicated to facilitating this process by offering a way of cross-silo cooperation using FedHP. We are still working on collecting larger-scale data and real-world hardware systems.
>
> `R2.2`: Although good results were reported on simulated data, more results on real data should be included to evaluate the effectiveness of the proposed method.
>
> `A2.2`: We provide additional comparisons on real data in Fig. 7$\sim$8. We are still working on collecting more data.
>
> `R2.3`: Why does the prompter lead to such a significant improvement, while the effect of the adaptor is not as pronounced? Please provide an in-depth analysis.
>
> `A2.3`: Thanks for the valuable comment. The prompter plays a crucial role in capturing hardware-specific characteristics and aligning inconsistent data distributions across clients. It directly addresses the heterogeneity rooted in the input data space, which is a key challenge in federated learning for SCI systems. By comparison, the adaptor, while important for efficient fine-tuning, has a more subtle effect on performance. Its primary role is to reduce communication costs in the FL system by allowing us to use pre-trained backbones and only communicate adaptors.
>
> `R2.4`:  The generalization to different hardware systems is crucial for deep learning based methods. The current form of this manuscript only reports on a small scale real dataset captured by several systems. A larger dataset captured by more systems is necessary to evaluate the method.
>
> `A2.4`: We kindly illustrate that current setup can consider a large amount of the real-hardware under two challenging scenarios.
> * Same distribution: As shown in Table 1, we sample non-overlapping masks from the same distribution for different clients. This simulates hardware perturbations or imperfections within the same system design.
> * Different distributions: In Table 2, we explore a more challenging scenario where each client has masks sampled from a specific distribution. This closely mimics diverse real-world scenarios, including hardware replacement or significant variations across institutions.
>
> These experiments, conducted on real hardware masks with noise (Fig. 9-11 in supplementary), demonstrate FedHP's ability to generalize across various hardware configurations. We appreciate the reviewer's insights on collecting larger dataset. This work laid the foundation for future extension by collecting a Snapshot Spectral Heterogeneous Dataset built upon multiple practical SCI systems for the first time. We are still working on collecting more data and real systems for better evaluation.

---

> > ### Comment · Reviewer_Jxiy · 2024-08-13
> >
> > Thanks for the explanations. Concerns like real-system evaluation cannot be addressed in such a short period. I keep my initial rating.

---

> > > ### Author Response · Authors · 2024-08-13
> > > **Response to Reviewer Jxiy**
> > >
> > > We appreciate the reviewer's recognition of our response and support for our work!

---

### Official Review · Reviewer_BdgG · 2024-07-13

**Soundness:** 3
**Presentation:** 3
**Contribution:** 3
**Rating:** 6
**Confidence:** 4

**Summary:**

The paper introduces FedHP, a reconstruction method for snapshot compressive imaging systems, which addresses the challenge of cross-hardware learning by proposing a federated learning approach. The key contribution lies in using a hardware-conditioned prompter to align data distributions across different hardware configurations, thereby enhancing the adaptability of pre-trained models without compromising data privacy.

**Strengths:**

1. The writing of the paper is good, making it easy to read and follow with clear arguments.
2. The problem defined in the paper is novel with a clear motivation, providing good inspiration for solving the issue of inconsistent device configurations in snapshot compressive imaging.
3. The proposed method is clear and the conclusions are relatively convincing. Overall, it is an interesting work.

**Weaknesses:**

1. There are some typos in the writing. For example, the caption of Figure 3 and the bold parts in the second row of Table 1 and the eighth row of Table 2 are confusing.
2. The proposed FedHP method is relatively straightforward and lacks deeper insights. Moreover, it does not show a significant performance improvement compared to FedAvg.
3. The experiments are not comprehensive enough. Given that this work aims to address the snapshot compressive imaging (SCI) problem, I suggest adding experiments to test the applicability of other SCI systems, such as Coded Aperture Compressive Temporal Imaging (CACTI).
4. There is a lack of sufficient real-world experiments. It would be beneficial to set up multiple independent SCI systems to test the algorithm's performance. Including reconstruction results obtained from these real-world systems is recommended.

**Questions:**

1. All the experiments in this paper are based on the SD-CASSI model. Can the same FedHP model be simultaneously applicable to both DD-CASSI and SD-CASSI architectures, which have significantly different designs?
2. Although the proposed method outperforms other algorithms in terms of performance metrics, there are still many artifacts in the reconstructed images. While I understand that this is maybe due to the precision issues of the CASSI system, it is crucial for evaluating the practical usability of the algorithm. Additionally, I am not sure whether the spectral accuracy of the reconstructed images is also optimal in statistical terms, which is vital for spectral imaging systems.
3. Furthermore, if possible, I hope the authors can also address the concerns I raised in the Weaknesses section.

**Limitations:**

Yes

---

> ### Author Rebuttal · Authors · 2024-08-06
>
> We much appreciate that the Reviewer `BdgG` provides valuable comments and finds the problem novel and the method convincing.
>
> `R1.1`: There are some typos in the writing. For example, the caption of Figure 3 and the bold parts in the second row of Table 1 and the eighth row of Table 2 are confusing.
>
> `A1.1`: We will correct the caption of Fig. 3 and the bold parts in Tables 1 and 2 to improve clarity. We will thoroughly proofread the manuscript to eliminate other errors.
>
> `R1.2`: The proposed FedHP method is relatively straightforward and lacks deeper insights. Moreover, it does not show a significant performance improvement compared to FedAvg.
>
> `A1.2`: FedHP offers insightful contributions:
> * It explicitly models hardware variations in computational imaging systems, which is non-trivial and addresses a key practical challenge.
> * The plug-and-play prompt network allows learnable hardware configurations, enabling hardware-software co-optimization.
> * The combination of prompt learning and adaptors reduces communication costs in federated learning systems.
>
> We empirically find that FedHP enables a consistent performance boost over FedAvg on different number clients, demonstrating the scalability of the proposed method. Considering that FedAvg is a strong baseline that even outperforms prevalent FL methods such as FedProx and SCAFFOLD, it is non-trivial to achieve performance boost over FedAvg.
>
> `R1.3`: I suggest adding experiments to test the applicability of other SCI systems, such as Coded Aperture Compressive Temporal Imaging (CACTI).
>
> | Metrics     | FedAvg     |      FedHP |
> |:-------------:|:-------------:|:--------------------------:|
> | PSNR      |  $27.35\pm1.22$    |  $27.87\pm0.89$   |
> | SSIM       | $0.9174\pm0.0046$ | $0.9192\pm0.0047$ |
>
> **Table T1. Comparison between FedAvg and FedHP on CACTI (Client:3)**
>
> `A1.3`: We launch additional experiments by applying FedHP to the real CACTI systems. The results in Table T1 above present a better reconstruction performance of FedHP over FedAvg baseline (under the same setting as Table 1),  demonstrating the generalizability ability of FedHP over a new SCI architecture. We will add the above discussion in the manuscript.
>
> Due to the limited rebuttal time and the heavy workload in deploying real CACTI systems, we can only obtain the intermediate results of both methods (e.g., $1.2\times 10^4$ out of $4\times 10^4$). We are committed to updating the latest results during the discussion stage.
>
> `R1.4`: There is a lack of sufficient real-world experiments. It would be beneficial to set up multiple independent SCI systems to test the algorithm's performance.
>
> `A1.4`: We kindly illustrate that all experiments are performed upon multiple independent real-world SCI systems. For both training and testing, we not only consider sampling non-overlapping masks from the same mask distribution, but also explore a challenging scenario where each client can have real masks sampled from a specific distribution, mimicking the diverse real-world scenarios including hardware perturbation and replacement (L27).
>
> `R1.5`: All the experiments in this paper are based on the SD-CASSI model. Can the same FedHP model be simultaneously applicable to both DD-CASSI and SD-CASSI architectures, which have significantly different designs?
>
> `A1.5`: It is challenging to apply FedHP simultaneously to both DD-CASSI and SD-CASSI architectures due to their different optical designs and forward models. DD-CASSI uses two dispersive elements and a coded aperture, while SD-CASSI uses a single dispersive element and a coded aperture, resulting in distinct measurement formation processes. These differences lead to incompatible forward models, making it difficult to unify them under a single reconstruction model. However, we agree this is an interesting direction for future research.
>
> `R1.6`: There are still many artifacts in the reconstructed images. While I understand that this is maybe due to the precision issues of the CASSI system, it is crucial for evaluating the practical usability of the algorithm.
>
> `A1.6`: Assessing the practical usability of SCI algorithms is inherently difficult due to the complex interplay of hardware limitations, reconstruction algorithms, and application-specific requirements. The proposed FedHP method itself is actually a step towards uncovering and tackling these practical challenges. By explicitly modeling hardware variations and enabling cross-hardware learning, FedHP aims to address real-world issues that arise when deploying SCI systems across multiple devices or institutions.
>
> `R1.7`: I am not sure whether the spectral accuracy of the reconstructed images is also optimal in statistical terms, which is vital for spectral imaging systems.
>
> | Methods | Scene 1 | Scene 2 | Scene 3 |
> |----------|----------|----------|----------|
> | FedAvg | 0.9816 | 0.8905| 0.8634 |
> | FedHP | **0.9901** | **0.9523** | **0.8851** |
>
> **Table T2. Spectral accuracy comparison on three testing scenes.**
>
> `A1.7`: We provide the spectral accuracy comparison in Fig.3, 5, and 6. Specifically, we randomly select a visually informative region (patch) on the hyperspectral images and compute the density values for a specific wavelength (i.e., dividing pixel values on this specific wavelength by the summarized pixel values along all the wavelengths). We statistically measure the spectral accuracy by computing the correlation of the density values between the prediction and the ground truth. FedHP enables better spectral accuracy by comparison.
>
> We further perform a statistical experiment on three simulation data, on which we randomly choose 10 spatial regions for the averaged correlation computation. As shown in Table T2 above, FedHP consistently enables a higher averaged correlation over FedAvg, indicating a better spectral accuracy. We will add the above discussions into the final version.

---

> > ### Comment · Reviewer_BdgG · 2024-08-13
> >
> > The author has addressed most of my concerns. I will keep my score unchanged.

---

> > > ### Author Response · Authors · 2024-08-13
> > > **Response to Reviewer BdgG**
> > >
> > > We appreciate the reviewer's approval for our response and recognition of our work!

---

### Decision · Program_Chairs · 2024-09-25

**Decision:**

Accept (poster)

**Comment:**

The paper presents a hardware focused method for performing snapshot compressive imaging in cases where differences in hardware are adjusted in the trained model. Federated learning is used towards this end. Four reviewers considered the paper, with two voting Weak Accept and two Borderline Accept. There was good engagement in the discussion period to indicate that they think the idea a good one and the results are potentially of interest. While the average score for this paper technically puts it on the threshold between accept and reject, the small variance between reviewers, and the fact that all consider the paper above threshold for accept, is a strong signal that it will contribute to the conference and is worth accepting.